# Triplets electrically turn on insulating lanthanide-doped nanoparticles

Zhongzheng Yu[1,4], Yunzhou Deng[1,4], Junzhi Ye[1,2,4], Lars van Turnhout[1], Tianjun Liu[1], Alasdair Tew[1], Rakesh Arul[1], Simon Dowland[1], Yuqi Sun[1], Xinjuan Li[3], Linjie Dai[1], Yang Lu[1], Caterina Ducati[3], Jeremy J. Baumberg[1], Richard H. Friend[1], Robert L. Z. Hoye[2] & Akshay Rao[1✉]

Insulating nanomaterials have large energy gaps and are only electrically accessible under extreme conditions, such as high-intensity radiation and high temperature, pressure or voltage[1,2]. Lanthanide-doped insulating nanoparticles (LnNPs) are widely studied owing to their exceptional luminescence properties, including bright, narrow-linewidth, non-blinking and non-bleaching emission in the second near-infrared (NIR-II) range[3,4]. However, it has not been possible to electrically generate excited states in these insulating nanomaterials under low biases and, therefore, not possible to fabricate optoelectronic devices from these systems. Here we report an electrical excitation pathway to obtain emission from LnNPs. By forming LnNP@organic molecule nanohybrids, in which the recombination of electrically injected charges on the organic molecule is followed by efficient triplet energy transfer (TET) to the LnNP, it is possible to turn on LnNPs under a low operating bias. We demonstrate this excitation pathway in light-emitting diodes (LEDs), with low turn-on voltages of about 5 V, very narrow electroluminescence (EL) spectra and a peak external quantum efficiency (EQE) greater than 0.6% in the NIR-II window[5]. Our LnNP-based LEDs (LnLEDs) also allow for widely tunable EL properties, by changing the type and concentration of lanthanide dopants. These results open up a new field of hybrid optoelectronic devices and provide new opportunities for the electrically driven excitation sources based on lanthanide nanomaterials for biomedical and optoelectronic applications.

LnNPs consist of an inorganic insulating host, typically fluorides or oxides such as $NaGd/Y/LuF_4$ with a large energy gap of approximately 8 eV (ref. 6), with lanthanide ions embedded in the host lattice. LnNPs have high photo and chemical stability in various environments and have narrow and tunable emission in the NIR-II range (1,000–1,700 nm). This is in contrast to semiconductor-based systems, such as NIR-II emissive organic dyes or semiconducting colloidal quantum dots (QDs), which show broad emission spectra in this region owing to homogeneous broadening. This has motivated research into the application of LnNPs in stimulated-emission depletion microscopy[7], deep-tissue theranostics[4,8–10], sensing[11] and optical communication[12]. However, as these systems are not semiconductors, they cannot be used to construct electrically driven devices, as can be done for colloidal QDs[13,14], metal halide perovskites[15–17] or organic semiconductors[18,19].

It has previously been shown that triplet excitons on organic molecules can couple to the f-f transitions in lanthanide ions and that this enables TET between organic molecules and LnNPs[20]. Organic dye sensitization has proved effective to enhance the emission of LnNPs[21–25]. Here we use molecular triplet excitons to mediate the function of electrically driven LnNP-based optoelectronic devices, using triplets to efficiently turn on these insulating materials. The first step in this process is to engineer the coupling between organic molecules and LnNPs. The inset of Fig. 1a shows a schematic of the LnNP. The as-prepared

LnNPs have oleic acid (OA) on the surface. However, OA is an insulating ligand, which cannot mediate electrical excitation. We therefore partially replace OA with 9-anthracenecarboxylic acid (9-ACA), a widely studied organic dye with a singlet energy of 3.2 eV and triplet energy around 1.8 eV (ref. 26). As shown in Fig. 1b, the triplet energy level of 9-ACA (ref. 27) can, in principle, allow for TET to the ladder-like energy levels of $Ln^{3+}$ ions (Ln = Nd, Yb, Er). These hybrid materials allow us to construct the first LnLEDs.

Figure 1a shows the device architecture of LnLEDs, consisting of glass/indium tin oxide (ITO)/poly(ethylenedioxythiophene):polystyrene sulfonate (PEDOT:PSS)/poly(4-butylphenyl-diphenylamine) (poly-TPD)/LnNP@9-ACA/1,3,5-tris(3-pyridyl-3-phenyl)benzene (TmPyPB)/lithium fluoride (LiF)/aluminium (Al). ITO and LiF/Al function as electrodes. PEDOT:PSS acts as hole injection layer. TmPyPB and poly-TPD serve as electron and hole transport layers (ETL and HTL), respectively. The LnNP@9-ACA nanohybrids serve as the light-emitting layer. Electrons and holes injected from the contacts travel through the charge transport layers and recombine on the 9-ACA ligands. This will lead to the formation of singlet and triplet excitons on 9-ACA in a 1:3 ratio as governed by the spin–statistics theorem. We note that triplet excitons can undergo efficient energy transfer to the $Ln^{3+}$ ions as shown in Fig. 1b and as experimentally demonstrated in Fig. 3. The $Ln^{3+}$ ions can then emit photons, leading to EL from the device. We keep the

[1]Cavendish Laboratory, University of Cambridge, Cambridge, UK. [2]Inorganic Chemistry Laboratory, University of Oxford, Oxford, UK. [3]Department of Materials Science and Metallurgy, University of Cambridge, Cambridge, UK. [4]These authors contributed equally: Zhongzheng Yu, Yunzhou Deng, Junzhi Ye. ✉e-mail: ar525@cam.ac.uk

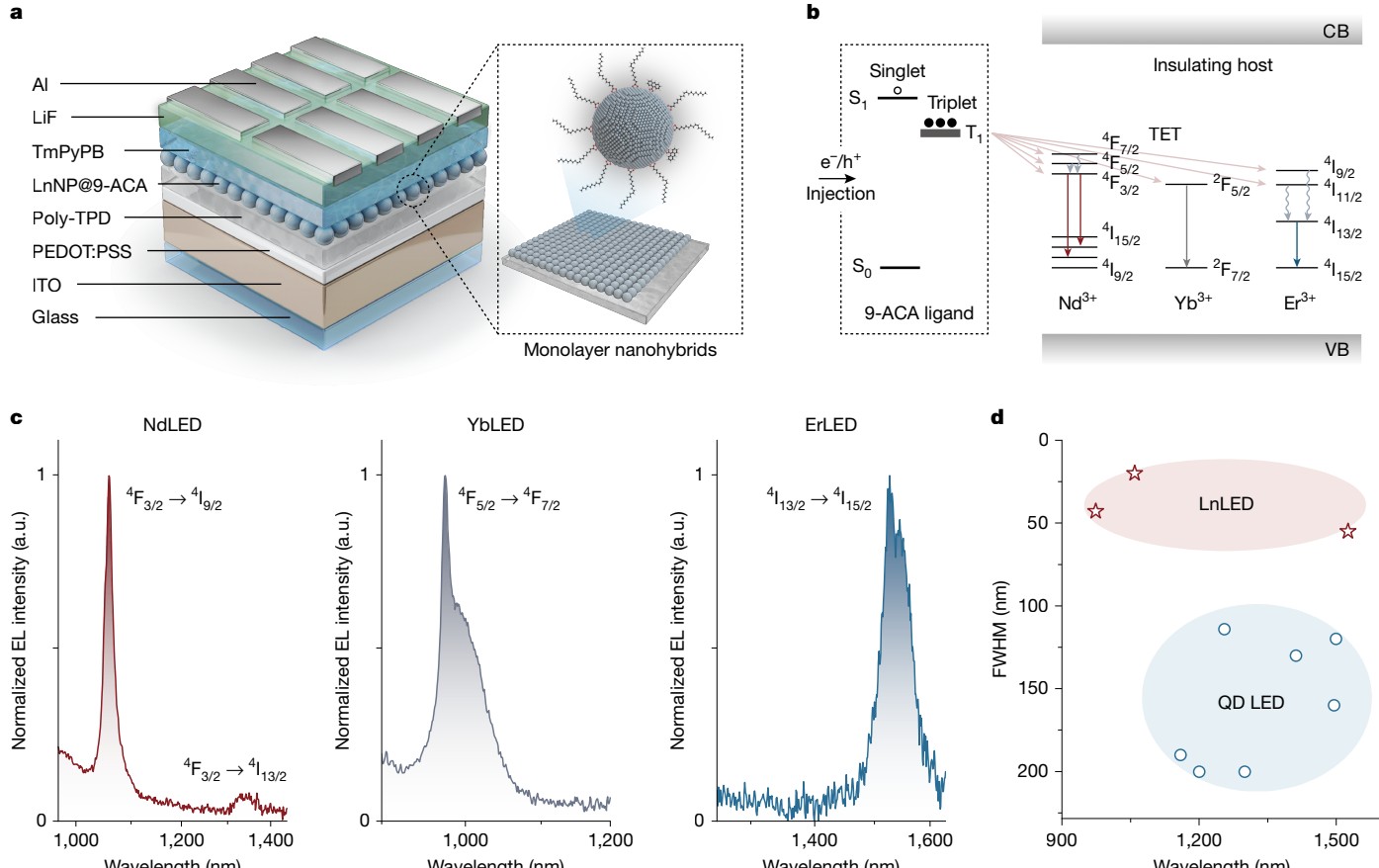

**Fig. 1 | Fabrication of LnNP-based NIR-II LEDs. a**, Schematic illustration of the device architecture of LnLEDs with a close-up schematic of LnNP@9-ACA nanohybrids. **b**, Simplified schematic showing electron and hole injection through organic molecules to turn on lanthanide ions in an insulating host lattice. CB, conduction band; VB, valence band. **c**, Normalized EL spectra of LnLEDs. a.u., arbitrary units. **d**, Reported FWHMs of the EL at different wavelengths from different types of LED, including LnLEDs and QD LEDs.

device architecture constant but vary the type of $Ln^{3+}$ ions doped into the LnNPs to achieve a range of EL emission from 1,000 to 1,533 nm.

Figure 1c shows the EL spectra obtained from the LnLEDs. The spectra are narrow and consistent with the main peaks of NIR-II photoluminescence (PL) spectra of LnNP@9-ACA nanohybrids under 350-nm photoexcitation. The full widths at half maximum (FWHMs) of LnLEDs EL spectra are calculated to be 20, 43 and 55 nm for Nd/Yb/ErLEDs, which are much lower than the FWHMs found in semiconducting QDs/bulk materials-based systems (FWHM normally above 150 nm)[5] (Fig. 1d). The large FWHM of QD LEDs, which is limited by homogeneous line broadening, creates complications for their use in optical communication and chemical/biomedical imaging/sensing applications. The narrow linewidths we achieve here, combined with the inherent ease of processing, flexibility, wide-area compatibility and potential low cost of organic–LnNP hybrids offers exciting possibilities for a new generation of light sources across the NIR-II range. A quantitative comparison of our LnLEDs and other NIR-II LEDs and laser diodes is included in Supplementary Tables 1 and 2.

To obtain high-quality NIR-II light-emitting layers, we synthesized uniform and ultrasmall (<10 nm) LnNPs. Transmission electron microscopy (TEM) images show that all of the LnNPs had good size monodispersity, with an average size of around 6 nm (Supplementary Fig. 1). The dopant ratio of fluorescent $Ln^{3+}$ (Ln = Nd, Yb, Er) ions has been fixed to 20 mol% in the form of $NaGd_{0.8}F_4{:}Ln_{0.2}$, which will be subsequently referred to as NdNPs, YbNPs and ErNPs, respectively. This dopant ratio guarantees that enough fluorescent $Ln^{3+}$ ions receive energy transferred from organic molecules and a fair comparison of energy transfer efficiencies among different $Ln^{3+}$ ions, while avoiding severe cross-relaxation to maintain a relatively high NIR-II fluorescence[28]. The high-resolution TEM images and X-ray diffraction (XRD) patterns show that these LnNPs are hexagonal phase (Supplementary Figs. 2 and 3).

As shown in Fig. 2a, the LnNPs show weak and narrow absorption peaks, which is one of the key limitations of LnNPs for various applications. Coupling 9-ACA onto the surface of LnNPs endows LnNP@9-ACA nanohybrids with strong absorption in the ultraviolet range (Fig. 2b). The absorption of these nanohybrids is hence dominated by organic molecules and overcomes the aforementioned limitation of LnNPs. LnNP@9-ACA nanohybrids also show a 5-nm redshift of absorption compared with pure 9-ACA owing to the coupling between organic molecules and LnNPs (Supplementary Fig. 4). Investigations of the ligand exchange process using Fourier-transform infrared (FTIR) spectroscopy and corresponding density functional theory (DFT) simulations (see Fig. 2c,d and Supplementary Figs. 5–7 for details) indicate that the 9-ACA preferentially binds to the $Ln^{3+}$ ion site on the surface of the LnNPs, in contrast to the OA, which also binds to the $Na^+$ sites. DFT-predicted FTIR spectra of 9-ACA bonded to $Gd^{3+}$ reproduces the experimentally observed spectrum, whereas 9-ACA bonded to $Na^+$ does not, and introduces peaks at 1,600 $cm^{-1}$, which are not observed (vertical lines in Fig. 2c). DFT-predicted FTIR spectra for OA show peaks shared at 1,450 and 1,590 $cm^{-1}$ for OA bonded to $Na^+$ or $Gd^{3+}$ (vertical lines in Fig. 2d). On the basis of the FTIR data, we estimate the replacement ratios of 9-ACA on different LnNPs to be 6.8%, 1.0% and 3.6% for NdNPs, YbNPs and ErNPs, respectively (Extended Data Fig. 1 and Supplementary Table 3). The important point here is the preferential binding of 9-ACA to the $Ln^{3+}$ ion sites, which will promote efficient energy transfer.

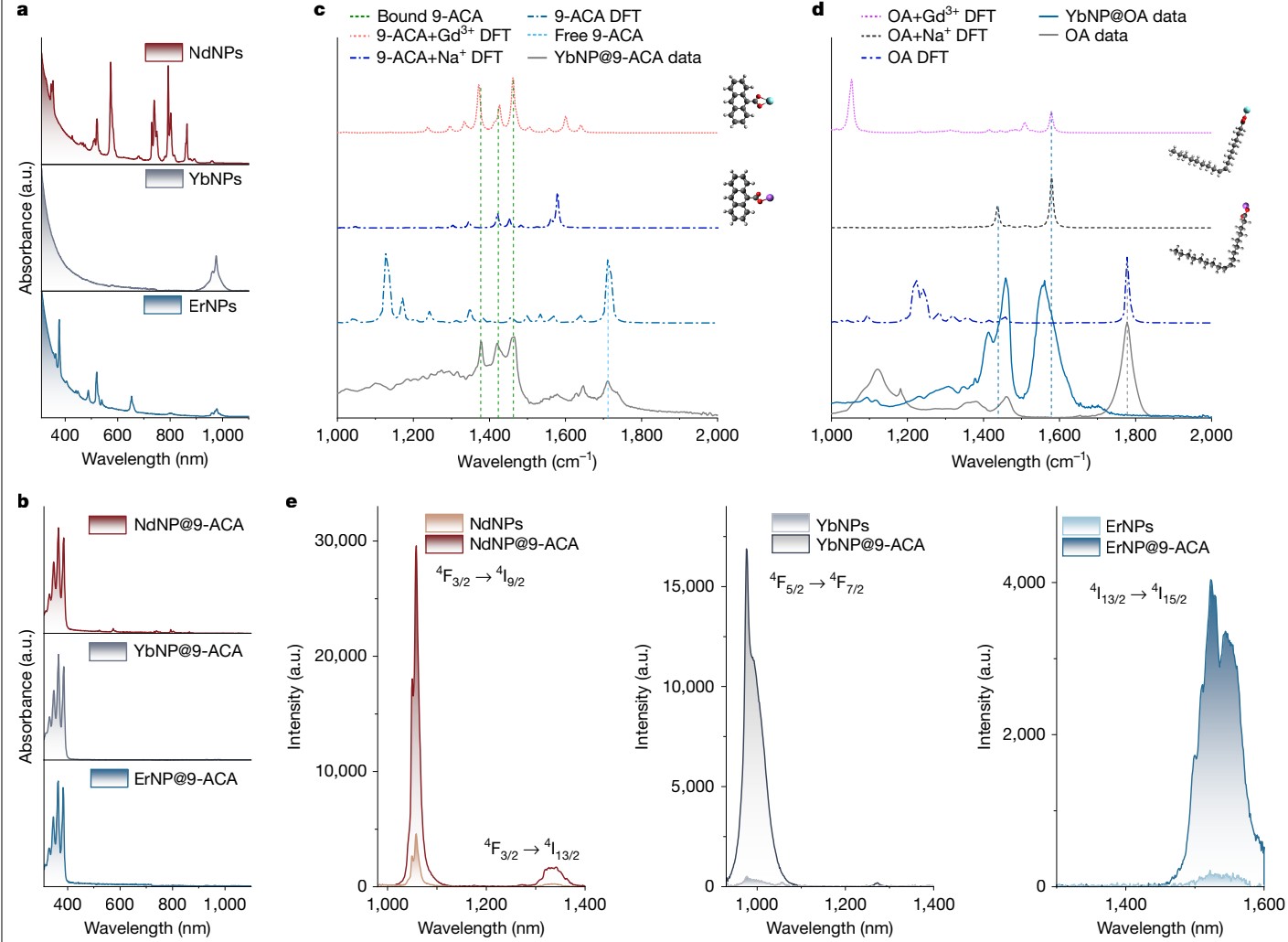

**Fig. 2 | Characterization of the LnNP@9-ACA nanohybrid system.**
**a**,**b**, Absorption spectra of LnNPs (**a**) and LnNP@9-ACA nanohybrids (**b**).
**c**, DFT-simulated FTIR spectra of free 9-ACA molecules, bound 9-ACA molecules to $Gd^{3+}$ and $Na^+$ ions and experimental data of YbNP@9-ACA nanohybrids. **d**, DFT-simulated FTIR spectra of free OA molecules, bound OA molecules to $Gd^{3+}$ and $Na^+$ ions and experimental data of OA-capped YbNPs. **e**, Comparison of NIR-II emission between LnNPs and LnNP@9-ACA nanohybrids under the excitation of a 350-nm lamp (concentration 20 mg ml$^{-1}$).

Ligand exchange is a dynamic process and can be influenced by numerous factors. We find that the ligand exchange rate is first increased by prolonging the reaction time and then reaches a plateau, by monitoring the absorbance change of YbNP@9-ACA nanohybrids and PL excitation spectra (Supplementary Fig. 8). We observe that the energy transfer efficiency is not greatly influenced by the ligand exchange rate when the ligand exchange process reaches an equilibrium (Supplementary Figs. 8d–f and 9). We also note that the short distance between 9-ACA and the surface of the LnNPs, linked by the carboxyl group, should allow for efficient TET, as this process is considered to be a Dexter-type energy transfer process. As well as the TET, energy transfer from the singlet state of the 9-ACA to the $Ln^{3+}$ ions is also possible by means of Förster resonance energy transfer (FRET), although the low absorption cross-section of the $Ln^{3+}$ ions and poor spectral overlap with the 9-ACA blue emission make this process inefficient[29].

As shown in Fig. 2e, the coupling of organic molecules leads to a notable enhancement of the NIR-II emission under ultraviolet excitation, achieving a large Stokes shift. The LnNP@9-ACA nanohybrids show 6.6-fold, 34.1-fold and 23.6-fold enhancement in NIR-II PL compared with NdNPs, YbNPs and ErNPs, respectively. The NIR photoluminescence quantum efficiencies (PLQEs) of LnNPs and LnNP@9-ACA nanohybrids are measured in Supplementary Table 4. Tuning the doping ratio of $Ln^{3+}$ is a straightforward and effective approach to enhance the fluorescent performance of LnNPs. Increasing the doping ratio of $Yb^{3+}$ would substantially enhance the downconversion intensities for both YbNPs and YbNP@9-ACA nanohybrids (Supplementary Fig. 10), for which cross-relaxation between $Yb^{3+}$ ions is not a notable loss, unlike in $Er^{3+}$ and $Nd^{3+}$. The ratios of several peaks in the NIR-II EL have changed compared with the PL spectra, indicating distinct energy transfer mechanisms under photoexcitation and electroexcitation for LnNP@9-ACA nanohybrids. To study the energy transfer mechanisms, we further perform steady-state PL, PL decay and transient absorption measurements. Owing to the different amounts of attached 9-ACA in the nanohybrids, we cannot directly compare the intensity of the visible PL to determine the efficiency of the energy transfer (Fig. 3a). PLQE measurements show that bound 9-ACA molecules on LnNPs have markedly decreased PLQE compared with pristine 9-ACA (Supplementary Fig. 11).

Figure 3b shows time-correlated single photon counting (TCSPC) results for Nd/Yb/ErNP@9-ACA nanohybrids. The lifetime of emission from 9-ACA decreased from 7.97 ns for the pristine ligand to 3.14, 6.77 and 6.94 ns for 9-ACA molecules bound onto Nd/Yb/ErNPs, respectively. Owing to the inevitable presence of free 9-ACA molecules in the nanohybrid solution samples, TCSPC results can merely indicate that energy transfer occurs within the nanohybrids but cannot provide a quantitative analysis of energy transfer efficiency. Hence, pump–probe spectroscopy is applied to reveal the dynamics of photoexcitation and

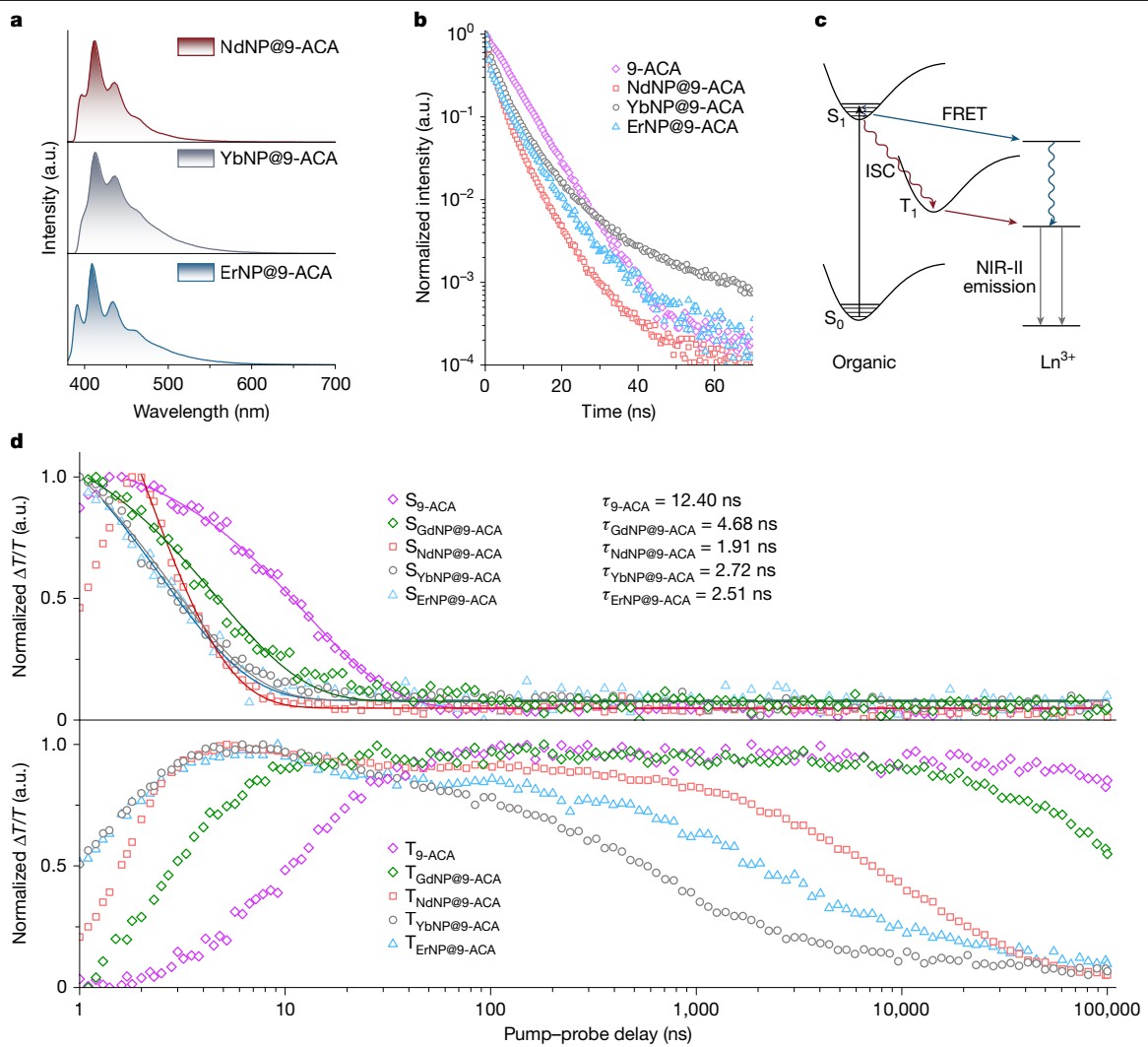

**Fig. 3 | Energy transfer from organic molecules to LnNPs. a**, Visible emission spectra of LnNP@9-ACA nanohybrids. **b**, TCSPC measurements of pristine 9-ACA and LnNP@9-ACA nanohybrids under 405-nm laser excitation. **c**, Schematic demonstration of the accelerated triplet generation by fast ISC when coupling Ln³⁺ ions and their corresponding energy transfer pathways involving both FRET and TET from organic molecules to Ln³⁺ ions. **d**, Kinetics of singlet decay, triplet growth and decay in pristine 9-ACA molecules and molecules attached to different types of LnNPs.

energy transfer between 9-ACA and LnNPs. As shown in Fig. 3d, the $S_1$ states of 9-ACA on NdNP, YbNP and ErNP decay with time constants of 1.91, 2.72 and 2.51 ns, respectively, which is much shorter than the singlet of the pristine 9-ACA and GdNP@9-ACA nanohybrids, which show decay time constants of 12.40 and 4.68 ns, respectively. The rise time constants of the triplet excitons ($T_1$) are measured to be 1.93, 1.41 and 1.39 ns for Nd/Yb/ErNP@9-ACA nanohybrids, respectively. By contrast, the pristine 9-ACA presents a triplet rise time of more than 12.97 ns (Extended Data Fig. 2). This indicates that the singlets of 9-ACA coupled onto LnNPs all undergo rapid intersystem crossing (ISC) and LnNPs increase the rate of the ISC (Fig. 3c), consistent with previous results on the coupling of triplet excitons to the unpaired spin of the doped lanthanide ions[20].

Figure 3d also shows much faster decays of the $T_1$ state in Nd/Yb/ErNP@9-ACA nanohybrids compared with the $T_1$ state decay of pristine 9-ACA and GdNP@9-ACA nanohybrids (see also Supplementary Fig. 12). These faster decays are caused by the energy transfer from 9-ACA to the $^2F_{5/2}$ level of YbNPs, $^4F_{3/2}$ level of NdNPs and $^4I_{11/2}$ levels of ErNPs, as GdNPs do not have energy levels available for energy transfer (Fig. 1b). The efficiencies of TET are calculated to be 98.8%, 99.8% and 99.4% for Nd/Yb/ErNP@9-ACA nanohybrids, respectively, based on the quenching of the triplet lifetime compared with GdNP@9-ACA (Extended Data Fig. 2). Both the efficient TET and the less efficient singlet FRET will contribute to the NIR emission of different LnNPs under light excitation. We measured the NIR PL intensities of YbNP@9-ACA nanohybrids under the $O_2$-free and air exposure conditions (Supplementary Fig. 13). The NIR PL in air was quenched by 53.3%, which is consistent with $O_2$ quenching the triplet state of 9-ACA. This further suggests that energy transfer from 9-ACA to LnNPs is mainly mediated by TET. We note three important points about the TET to LnNPs. First, the lifetime of the triplet exciton on 9-ACA is longer than 300 μs, whereas the TET times are on the order of several microseconds. This means that the triplets on the ligands provide a long-lived state from which energy transfer can occur, with few competing kinetic processes, enabling highly efficient transfer (>98%). Second, in Dexter energy transfer, which is the mechanism for TET to LnNPs, the spectral overlap between the triplet phosphorescence spectrum and acceptor absorption spectrum is an important factor[20]. The phosphorescence spectrum of 9-ACA, which has been reported previously, is broad (roughly 1.3–1.9 eV)[27], overlapping with numerous levels within the Ln³⁺ and allowing for TET. Third, although singlet transfer from the $S_1$ state of 9-ACA to the Ln³⁺ levels is also possible, the short lifetime of the pristine singlet (12.4 ns), which is further reduced by ISC when attached to the Ln³⁺ ions with unpaired spins (<5 ns), lowers the efficiency of this pathway, especially given the

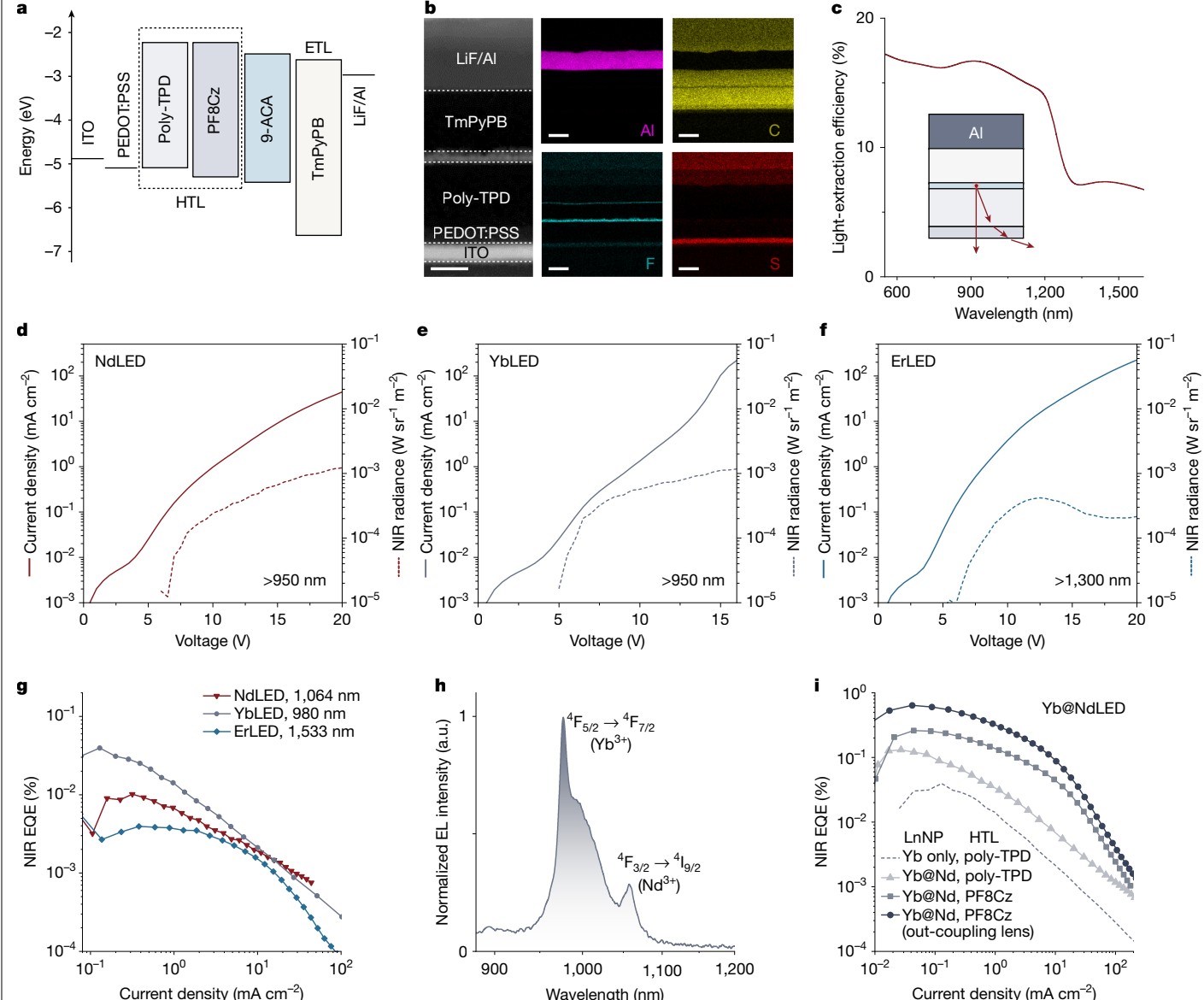

**Fig. 4 | Characterization and optimization of NIR-II LnLEDs. a**, Energy band diagram of LnLEDs. **b**, Cross-sectional HAADF STEM image of the YbLED and corresponding element mapping of different layers. Scale bars, 50 nm. **c**, Simulated light-extraction efficiency of the LnLED as a function of emitting wavelength in the NIR range. **d–f**, Current density and NIR radiance (measured with corresponding long-pass filters) versus voltage for different LnLEDs. **g**, NIR EQEs of the NdLED/YbLED/ErLED versus current densities. **h**, Normalized EL spectrum of the Yb@NdLED. **i**, NIR EQE enhancement by using core–shell Yb@Nd LnNPs and optimizing the device structure.

poor oscillator strength of the $Ln^{3+}$ ions. The TET pathway is thus the dominant energy transfer pathway and key to enabling the fabrication of optoelectronic devices.

We now turn to the structural characterization of LnLEDs. A schematic of the device structure with flat-band energy levels is shown in Fig. 4a. The device structure is designed to enable charge injection into 9-ACA, leading to exciton formation and subsequent energy transfer to the LnNP. The high-angle annular dark-field (HAADF) scanning transmission electron microscopy (STEM) image of the cross-section of the YbLEDs shows the multilayer structure of the hybrid device. The corresponding elemental mapping of the cross-section of the YbLEDs demonstrates a uniform distribution of Yb and Gd elements within the light-emitting layer (Fig. 4b and Supplementary Fig. 14), which is consistent with the scanning electron microscope results of LnNP@9-ACA nanohybrids on PEDOT:PSS/ITO substrates spin-coated at different rotation speeds (Supplementary Fig. 15). Grazing-incidence wide-angle

X-ray scattering (GIWAXS) measurements confirm that LnNP@OA and LnNP@9-ACA films on poly-TPD/PEDOT:PSS/ITO do not form a super-lattice (Supplementary Fig. 16), with either OA or 9-ACA ligands. The thicknesses of the layers are 25 nm (ITO), 20 nm (PEDOT:PSS), 80 nm (poly-TPD), 15 nm (YbNP@9-ACA), 80 nm (TmPyPB) and 100 nm (LiF/Al), respectively (Fig. 4b). These thicknesses are chosen to enable a relatively high light-extraction efficiency from the emitting layer of the LnLEDs, in the NIR spectral region, as shown in Fig. 4c (see Extended Data Fig. 3 for details).

The NIR-II EL spectra of Nd/Yb/ErLEDs show sharp peaks centred at 1,058, 976 and 1,533 nm, respectively. No shifts in peak emission wavelength were observed under varying driving voltages (Supplementary Fig. 17). Owing to the ladder-like energy levels of LnNPs, there are several peaks for Nd/ErLEDs. The EL of the presented LnLEDs also involves visible-range emission (Supplementary Fig. 18). We assign the blue EL to poly-TPD HTL. The red emission arises from the interface of directly

contacted HTL and ETL in the voids of the monolayer LnNP@9-ACA nanohybrids[30]. No clear EL from 9-ACA is observed, confirming the efficient ISC and TET in the LnNP@9-ACA nanohybrids. The visible EL features suggest electron leakage from ETL or LnNP@9-ACA to poly-TPD, which acts as an efficiency loss channel for the EQEs of LnLEDs.

The current density–voltage–radiance curves of these LnLEDs are shown in Fig. 4g–i. The turn-on voltages for LnLEDs, defined by the voltage corresponding to the minimum measurable radiance in our set-up (0.01 mW sr$^{-1}$ m$^{-2}$; see Supplementary Fig. 19), are all around 5 V. The LnLEDs can endure high voltages up to 15 V. As most of the triplets in organic molecules have been transferred to the robust LnNPs, the triplet-induced degradation in LnLEDs could be suppressed. This could allow the LnLEDs to function under high voltages. The peak radiances of the Nd/Yb/ErLEDs are 1.2, 1.2 and 0.4 mW sr$^{-1}$ m$^{-2}$, respectively (Fig. 4d–f). The peak EQEs of the Nd/Yb/ErLEDs in the NIR regime are around 0.01%, 0.04% and 0.004%, respectively (Fig. 4g).

The moderate EQEs are limited by the PLQE of the highly doped core-only LnNPs, charge leakage across the emitting layer (indicated by Supplementary Fig. 18) and the decreased light-extraction efficiency in the NIR-II range (Fig. 4c). To further boost the NIR EQEs of LnLEDs, we fabricated core–shell Yb@Nd LnNPs in the form of NaGd$_{0.8}$F$_4$:Yb$_{0.2}$@NaGd$_{0.4}$F$_4$:Nd$_{0.6}$ to substantially increase the PLQE of Yb@Nd@9-ACA nanohybrids to 3% under 375-nm excitation (Extended Data Fig. 4). Figure 4i compares the NIR EQEs of the Yb@NdLEDs that use this core–shell configuration, together with further optimized HTL with better hole-injection and electron-blocking properties[14] and a light out-coupling half-ball lens on substrate. These strategies can suppress the efficiency losses in LnLEDs, eventually boosting the peak NIR EQE of Yb@NdLEDs to greater than 0.6%. We note that the peak EQEs of the LnLEDs are higher than most organic LEDs (OLEDs) emitting above 1,000 nm (ref. 31) (Supplementary Table 1). The EL spectra show a higher Nd/Yb peak intensity ratio of 0.29 than the PL spectra of 0.22 by the direct excitation of Yb@Nd NPs (Fig. 4h and Extended Data Fig. 4), indicating that electrical excitation prefers active surface Ln$^{3+}$ ions and the energy transfer from Nd$^{3+}$ to Yb$^{3+}$ is less efficient than that under light excitation.

We fabricate control LEDs using 9-ACA and NdNP@OA as the emissive layers, respectively, using the same solution-processed method (Supplementary Fig. 20). No emission is detected from the OA-capped NdLEDs. We measure a peak EQE of 0.4% for the pure 9-ACA (without host matrix)-based OLED emitting in the visible range (Supplementary Fig. 20c). This indicates a relatively low electrical-excitation efficiency of the 9-ACA molecule, which could be caused by leakage of charge carriers or interfacial quenching of singlet excitons. These control experiments prove that the molecular antennas are crucial to turn on insulating LnNPs under low voltages.

In summary, this work establishes triplet-mediated electrical excitation as a method to turn on insulating lanthanide nanomaterials, by harvesting the energy from 'dark' molecular triplet excitons at low voltages. Using this, we have given the first proof-of-concept demonstration of LnLEDs. These LnLEDs represent the spectrally narrowest NIR-II EL reported so far, with a tolerance for driving voltages of more than 15 V. Our results reveal some of the key energy loss channels that limit this new class of LnLEDs at present, especially the use of monolayers of the nanohybrids and relatively low replacement ratio of 9-ACA on the surface of LnNPs (<10%). This limits the brightness of LnLEDs to a value much lower than that of QD LEDs. But the results also point to strategies to improve these devices in the future. For instance, materials and device optimization will allow for the development of better organic ligands and hosts matrices with optimized charge transport and recombination properties. This effort will be aided by experience built up in the OLED community over several decades. On the emitter side, we note that the PLQEs of ultrasmall LnNPs with high dopant concentration used in this study are very modest. These modest PLQEs serve as another loss channel and cap the efficiency of these first devices

(EQE >0.6%). In the future, tuning the doping ratio and doping types of Ln$^{3+}$, as well as control of particle size, will boost the NIR-II PLQEs[32,33], thus allowing for higher EQEs. For instance, it has been shown that PLQEs >50% can be achieved for Er$^{3+}$ emission at 1,530 nm in LnNPs[32]. Future experimental and theoretical work will thus be required to enhance the brightness and operational stability of LnLEDs (Extended Data Fig. 4). The general method we establish here can be applied to numerous organic molecules and various LnNPs, providing a new route to turn on insulating materials under low applied biases. This opens a new field for the design and fabrication of hybrid LEDs and also other electrically pumped devices, such as lasers, with huge potential in biomedical theranostics, optogenetics and optical communication.

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

## Methods

### Materials

Gadolinium(III) acetate hydrate (99.9%), neodymium(III) acetate hydrate (99.9%), ytterbium(III) acetate tetrahydrate (99.9%), erbium(III) acetate hydrate (99.9%), holmium(III) acetate hydrate, sodium hydroxide (NaOH, pellets, >98%), ammonium fluoride ($NH_4F$, 99%), OA (technical grade, 90%), 1-octadecene (technical grade, 90%), 9-ACA (99%) and other anhydrous solvents were purchased from Sigma-Aldrich. Unless stated otherwise, all chemicals were used as received without further purification.

### Synthesis of $NaGd_{0.8}F_4$:$Ln_{0.2}$ (Ln = Nd, Yb, Er, Ho) nanoparticles

The $NaLnF_4$ nanoparticles were synthesized by means of a modified co-precipitation method. 0.8 mmol $Gd(CH_3CO_2)_3$ and 0.2 mmol $Ln(CH_3CO_2)_3$ were dissolved in 6 ml OA and 14 ml 1-octadecene in a 50-ml three-neck glass flask. The whole reaction is conducted under the protection of inert gas ($N_2$). The reaction precursor was first heated at 140 °C in an oil bath and maintained for 1 h and then cooled to room temperature. 0.07 g NaOH and 0.104 g $NH_4F$ were dissolved in 7 ml of methanol and injected into the precursor and heated to 70 °C and maintained for 45 min to get rid of the methanol. The flask was then transferred to a heating mantle and the reaction temperature increased to 300 °C and maintained at this temperature for 40 min for nanocrystal growth. After that, the solution was cooled to room temperature and centrifuged at 7,500 rpm for 5 min after adding ethanol as antisolvent. The resulting LnNPs were dissolved in hexane for further purification, ligand exchange or storage.

### Ligand exchange to form LnNP@9-ACA nanohybrids

Ligand exchange of the OA-capped LnNPs was carried out after several washes by ethanol and dissolved in anhydrous hexane solvent. As 9-ACA has a low solubility in hexane, tetrahydrofuran was added to enhance the solubility in a mixture solution with a v/v ratio of 9:1. 1 ml of LnNPs in hexane (25 mg ml$^{-1}$) was mixed with 1 ml of 9-ACA in the mixture solution (1 mg ml$^{-1}$) in a sample vial and the ligand exchange was left overnight. The mixture solution was mixed with ethanol and centrifuged at 10,000 rpm for 5 min. The nanohybrids were obtained as precipitation and re-dissolved in hexane or octane for further use. The calculation of the ligand replacement ratio is shown in Supplementary Note 1.

### Sample characterization

XRD measurements were carried out on X-Ray Bruker D8 under an ambient environment with Cu Kα radiation ($\lambda$ = 1.5406 A). FTIR spectra were measured using attenuated total reflectance infrared spectroscopy (Shimadzu IRTracer 100). Samples were drop-cast into thin films onto gold-coated Si wafer substrates (thermally evaporated 100 nm Au, 5 nm Cr adhesion layer) before being pressed onto the attenuated total reflectance crystal. A total of 25 scans were averaged, with Happ-Genzel apodization. Spectra were subtracted from control samples to identify unique peaks associated with OA and 9-ACA to determine the degree of ligand replacement and surface coverage. PL spectra were measured on an FLS1000 PL spectrometer. The GIWAXS measurements were done in Diamond Light Source, Beamline I07, using an X-ray energy of 10 keV and Pilatus 2M detector, under He flow.

### DFT simulations

All infrared spectra were assigned with DFT simulations in Gaussian 09 of organic molecules attached to Na(I), Gd(III) and Y(III) atoms. DFT calculations were conducted on the Cambridge supercomputer. The B3LYP functional was used, with main group elements (C, H, O, Na) represented by the 6-31+G(d,p) basis set and the lanthanide elements (Gd, Y) by the MWB28 effective core potential. The high-spin version of the complexes was always calculated, with an ultrafine grid and quadratically convergent SCF optimization.

### Device fabrication

ITO/glass substrates (ITOTEK) were cleaned by ultrasonication in acetone and isopropanol for 15 min each and dried by blowing $N_2$ gas on the surface. The substrates were subsequently $O_2$-plasma treated for 10 min. The PEDOT:PSS solution (Heraeus Clevios P VP Al 4083) was spin-coated at 6,000 rpm for 30 s (4,000 rpm s$^{-1}$ acceleration) and then annealed at 145 °C for 20 min with $N_2$ gas flowing on the top surface. Then the substrates were transferred into a $N_2$-filled glovebox for the rest of the processing. The hole-transport materials poly-TPD (Sigma-Aldrich) or poly((9,9-dioctylfluorenyl-2,7-diyl)-*alt*-(9-(2-ethylhexyl)-carbazole-3,6-diyl)) (PF8Cz, Volt-Amp Optoelectronics Tech. Co., Ltd.) were dissolved in chlorobenzene (15 mg ml$^{-1}$) and spin-coated over the PEDOT:PSS at 1,000 rpm for 30 s and annealed at 150 °C for 20 min. The LnNP@9-ACA nanohybrids were subsequently deposited at 2,000 rpm for 30 s. The TmPyPB layer (Ossila) was thermally evaporated over the LnNP@9-ACA nanohybrids under a base pressure lower than $1 \times 10^{-6}$ mbar, followed by 1 nm LiF (Sigma-Aldrich) and 100 nm Al. The devices were then encapsulated by using glass slides with an ultraviolet-curable epoxy. For the device with external light out-coupling structure, an 8-mm-diameter half-ball lens (Edmund Optics) was attached onto the substrate through index-matching ultraviolet-curable epoxy.

### LED device characterization

Current density–voltage ($J$–$V$) characteristics were measured using a Keithley 2400 SourceMeter unit. A calibrated germanium power meter with known spectral responsivity (A W$^{-1}$ nm$^{-1}$) was used to characterize the photon flux and radiance of the EL in the NIR region (see Supplementary Fig. 19 for the detector properties). The visible emission was filtered out by using a long-pass filter (950 nm for Nd and YbLEDs, 1,300 nm for ErLEDs) attached to the power meter. The detector head ($10 \times 10$ mm$^2$) was placed in the normal direction of the device plane ($2 \times 2$ mm$^2$). The distance between the detector and the LED (15 mm) was more than five times larger than the device lateral dimension to fulfil the spot light-source assumption for the goniometric measurement. The detailed calibration and radiance and EQE calculation is shown in Supplementary Note 2. The EL spectra of the devices were measured using a calibrated grating spectrometer (Andor SR-303i). The accuracy of the spectral data was cross-checked against a Labsphere CDS-610 spectrometer. EL spectra of LnLEDs were measured on an Andor iDus spectrograph with both Si array detector and InGaAs detector. For simulation of the light-extraction efficiency of LnLEDs, the optical parameters of the organic charge transport layer, Al, ITO and PEDOT:PSS are taken from previous reports[14,34,35] and fitted in the Ansys Lumerical software to extrapolate to the NIR region.

### STEM characterization

Cross-sectional sample lamellae were prepared for STEM using the FEI Helios NanoLab DualBeam focused ion beam (FIB)/scanning electron microscope. The lamellae were thinned to achieve electron transparency, with a thickness of less than 200 nm. The lamellae were promptly transferred to an FEI Tecnai Osiris 80-200 transmission electron microscope to minimize air exposure to less than 2 mins. The acceleration voltage for the electron beam was set to 200 kV. STEM energy-dispersive X-ray spectroscopy images were acquired with a beam current of 140 pA, a dwell time of 30 ms per pixel, a spatial sampling of 10 nm per pixel and a camera length of 115 mm. The STEM energy-dispersive X-ray spectroscopy data underwent denoising using principal component analysis and were subsequently processed using HyperSpy.

### TCSPC measurements

The LnNP@9-ACA nanohybrid solution was photoexcited using a 350-nm pulsed laser with a pulse width <200 ps, at a repetition rate of 40 MHz. Photons emitted from the sample were collected by a

silicon-based single-photon avalanche photodiode. The instrument response function has a lifetime of about 0.2 ns.

## Transient absorption spectroscopy

The pump beam used for the transient absorption spectroscopy was a 355-nm pulsed laser operating at 500 Hz. It was generated from the third harmonic configuration of Picolo (AOT-MOPA 25, Picosecond Nd:YVO$_4$ Laser 494 System, INNOLAS) and was triggered at desired time delays (pulse duration about 800 ps). The broadband white probe was provided by Disco (Leukos Laser, STM-2-UV) and the pump–probe decay was controlled electronically. The white light was split into two identical beams (probe and reference) by a 50/50 beam splitter. The reference beam was used to correct for any shot-to-shot fluctuations in the probe that would otherwise greatly increase the structured noise in the experiments. It was achieved by passing the reference beam through the sample, which did not interact with the pump. This arrangement allowed small signals with $\Delta T/T$ of around $10^{-5}$ to be measured. The transmitted probe and reference pulses were collected with a silicon dual-line array detector (Hamamatsu S8381-1024Q, spectrograph: Andor Shamrock SR303i-B) driven and read out by a custom-built board (Stresing Entwicklungsbüro). The spectra were taken with liquid samples inside a 1-mm-thick cuvette.

## Data availability

The data underlying all figures in the main text are publicly available from the University of Cambridge repository at https://doi.org/10.17863/CAM.120957 (ref. 36).

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

**Acknowledgements** This work was supported by a UK Research and Innovation (UKRI) Frontier Research Grant (EP/Y015584/1). Z.Y. acknowledges funding from UKRI Postdoctoral Individual Fellowships (grant reference EP/X023133X/1). Y.D. acknowledges funding from UKRI Postdoctoral Individual Fellowships (grant reference EP/Y02771X/1). J.Y. and R.L.Z.H. thank UKRI for funding through a Frontier Grant (no. EP/X022900/1), awarded through the 2021 European Research Council (ERC) Starting Grant scheme. J.Y. and R.L.Z.H. thank St. John's College, Oxford for support through the Large Grant scheme. R.L.Z.H. also thanks the Royal Academy of Engineering for support through the Senior Research Fellowships scheme (no. RCSRF2324-18-68). L.v.T. acknowledges funding from the Winton Programme for the Physics of Sustainability and from the Engineering and Physical Sciences Research Council (EPSRC). A.T. acknowledges support from the UKRI NanoDTC Cambridge EP/S022953/1. Y.S. acknowledges funding from UKRI (grant reference EP/S030638/1) and China Scholarship Council and Cambridge Trust Scholarship. R.A. acknowledges support from St. John's College, Cambridge, the Rutherford Foundation of the Royal Society Te Apārangi of New Zealand and the Winton Programme for the Physics of Sustainability. J.J.B. acknowledges support from the ERC under the Horizon 2020 research and innovation programme THOR (grant agreement no. 829067) and PICOFORCE (grant agreement no. 883703). Y.L. acknowledges funding from the EPSRC (EP/V06164X/1). The GIWAXS measurement was carried out with the support of Diamond Light Source, Beamline I07 (proposal SI32266).

**Author contributions** Z.Y. and A.R. conceived this work. Z.Y. performed nanoparticle synthesis, ligand exchange, purification and absorption measurement. Y.D. developed the final device structure of the LnLEDs, fabricated the LnLEDs in this work, set up the high-sensitivity NIR LED efficiency measurement system and analysed light out-coupling of the LnLEDs under the supervision of A.R. J.Y. made initial attempts on the LnLED fabrication and fabricated the control devices under the supervision of A.R. and R.L.Z.H. T.L. cross-checked the LnLEDs under the supervision of R.H.F. L.v.T., A.T. and Z.Y. carried out spectroscopic measurements. Y.D., J.Y., S.D. and Y.S. performed EL measurements. J.Y., L.v.T. and L.D. contributed to transient absorption experiments and data analysis. R.A. carried out FTIR measurement and theoretical calculations under the supervision of J.J.B. A.T. performed TEM measurement. X.L. performed HAADF STEM measurement for the range of LnLEDs under the supervision of C.D. Y.L. performed XRD measurement. Y.L. and T.L. performed GIWAXS measurement and Y.L. performed GIWAXS data plotting and analysis under the supervision of R.H.F. Z.Y. and A.R. wrote the manuscript. Z.Y. and A.R. supervised the project. All authors discussed the results and commented on the manuscript.

**Competing interests** The authors declare no competing interests.

**Additional information**
**Correspondence and requests for materials** should be addressed to Akshay Rao.

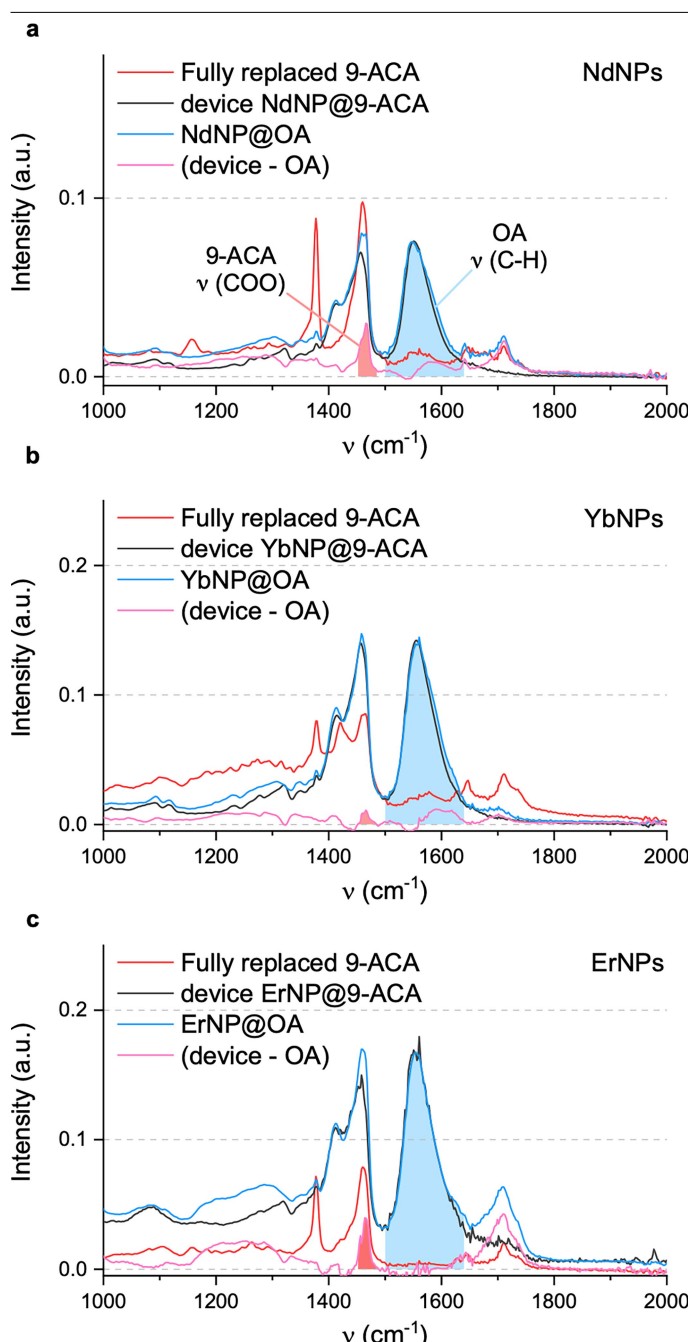

**Extended Data Fig. 1 | Fraction of OA replaced by 9-ACA.** Integrated area of OA (blue) and 9-ACA (red) used to calculate the fraction of 9-ACA replaced. **a**–**c**, Panels show the fully replaced 9-ACA LnNPs, the native OA-capped LnNPs and the partially replaced device LnNP@9-ACA nanohybrids, along with the subtraction spectrum of the device 9-ACA and native OA to reveal the 9-ACA coordinated for NdNPs (**a**), YbNPs (**b**) and ErNPs (**c**).

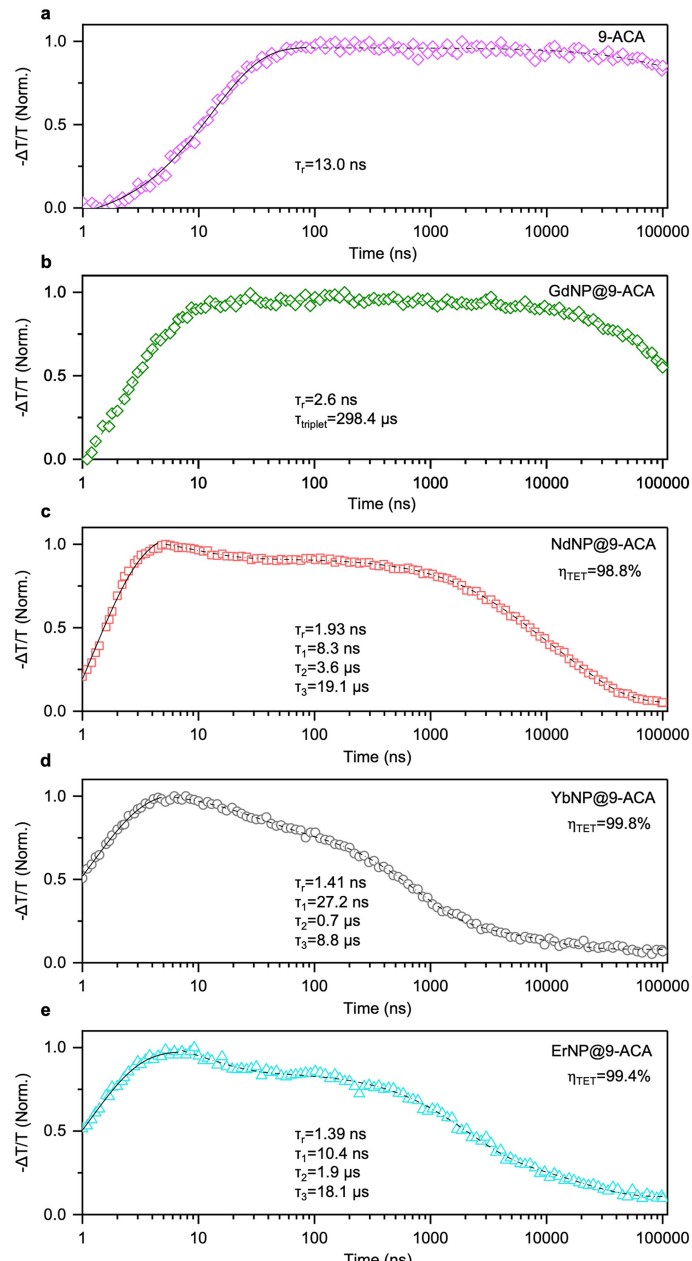

**Extended Data Fig. 2 | The calculation of efficiencies of TET from 9-ACA to LnNPs. a–e**, The fitted triplet rise and decay lifetime from the transient absorption measurement for pure 9-ACA (**a**), GdNP@9-ACA (**b**), NdNP@9-ACA (**c**), YbNP@9-ACA (**d**) and ErNP@9-ACA (**e**) nanohybrids. The Dexter triplet transfer efficiency is calculated using the formula $1 - (\tau_2/\tau_{triplet}) \times 100\%$.

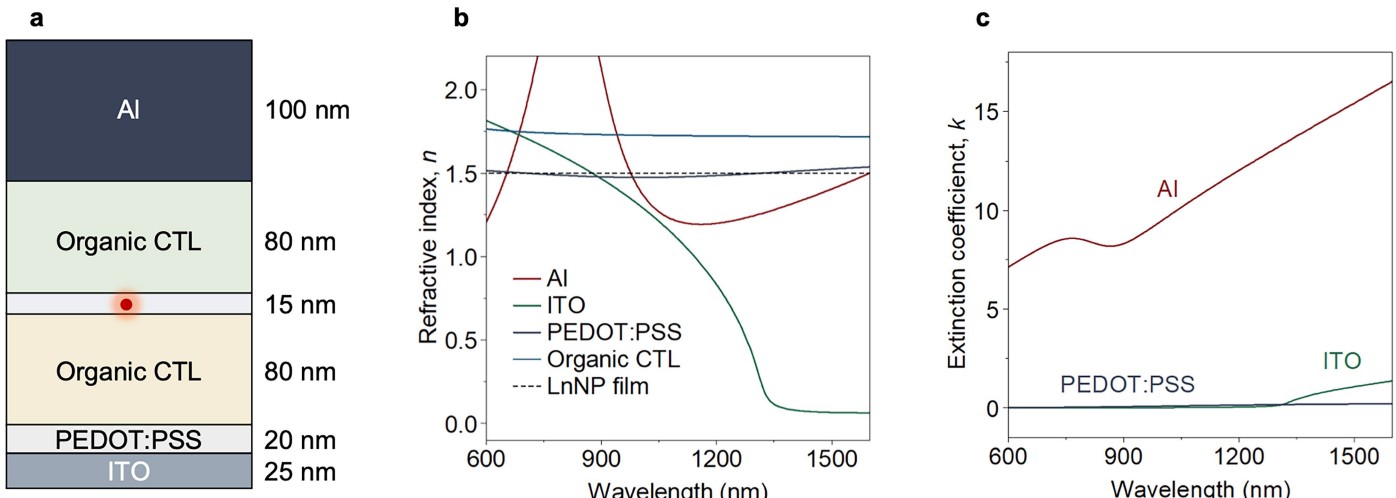

**Extended Data Fig. 3 | Optical simulation of the NIR-II LnLED structure. a**, Optical structure of the LnLED and the corresponding thickness of the functional layers. The simulation results are the sum of the results from three independent orthogonal dipoles placed in the central part of the emissive layer (indicated by the red dot). **b**, Refractive indices of the functional layers as a function of wavelength. **c**, Extinction coefficients of the functional layers as a function of wavelength. The organic charge transport layers (CTLs) and the LnNP film are treated as non-absorbing mediums in the simulation. The refractive index of the LnNP film is approximated as 1.6 considering the low packing density of the NPs and relatively low refractive index of fluoride matrix.

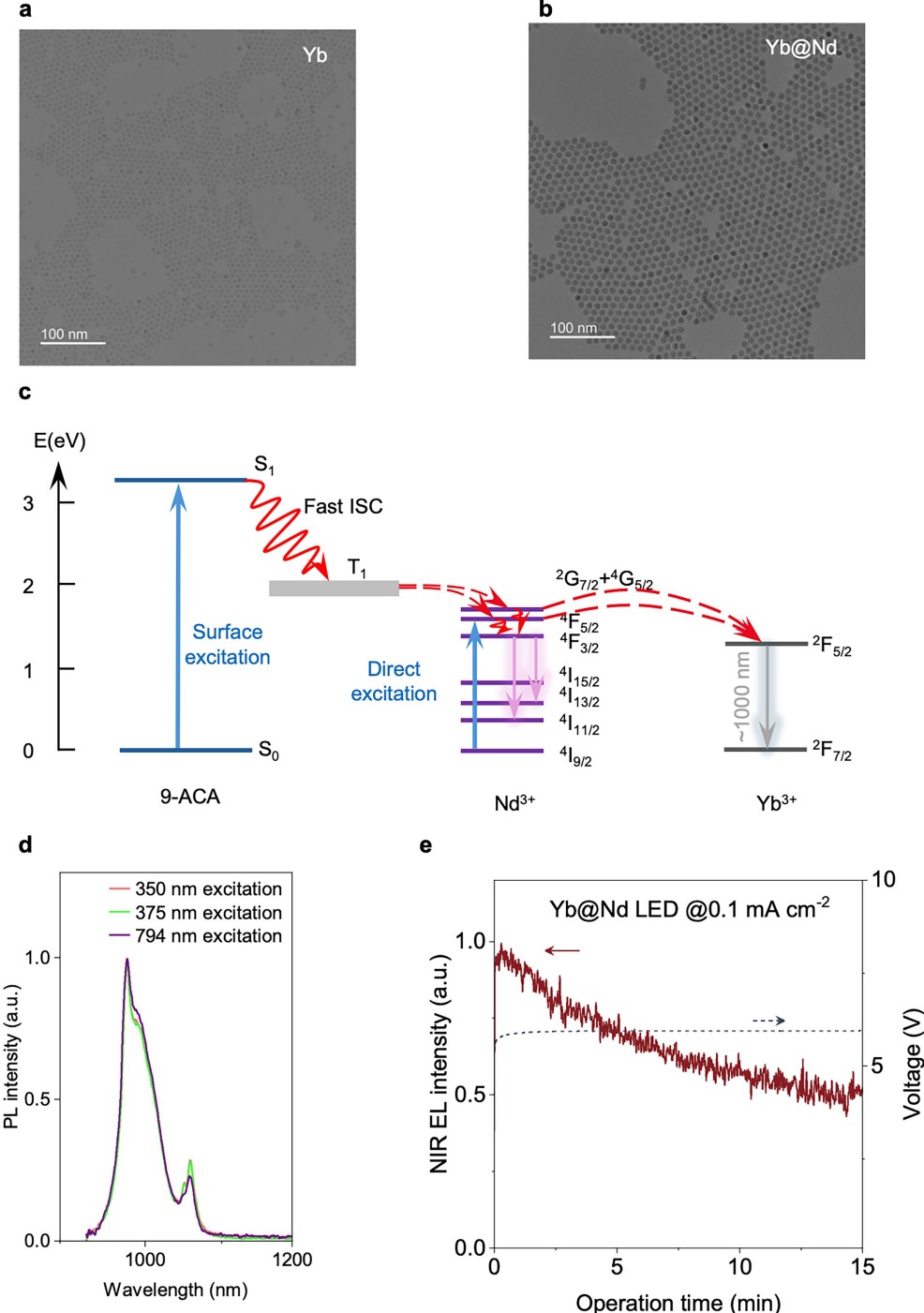

**Extended Data Fig. 4 | Characterization and energy transfer of the core–shell LnNPs system. a,b**, TEM images of Yb core LnNPs with an average size of about 6 nm (**a**) and Yb@Nd core–shell LnNPs with an average size of about 10 nm (**b**). **c**, Simplified energy transfer diagram from 9-ACA to $Nd^{3+}$ to $Yb^{3+}$ through surface excitation and from $Nd^{3+}$ to $Yb^{3+}$ through direct excitation of $Nd^{3+}$. **d**, Normalized PL spectra of Yb@Nd@9-ACA nanohybrids under different excitation wavelengths at 350, 375 and 794 nm. **e**, Operational stability of the Yb@Nd LED at constant current density.