## [Peer Review File · Nature]

Triplets electrically turn on insulating lanthanide-doped nanoparticles

Corresponding Author: Professor Akshay Rao

Version 0:

Reviewer comments:

Referee #1

(Remarks to the Author)

This work describes the first demonstration of electrically-pumped emission from Ln-doped nanoparticles, overcoming an important barrier in these materials. They successfully fabricated the LED devices by combining dye-sensitized LnNPs with robust architecture concepts used in other device types. They show relatively narrow emission at various shortwave infrared (SWIR) frequencies, depending on the Ln species. Beyond the electrical excitation, there are some other excellent results here, including (1) a dye-NP attachment strategy that allows for direct bonding of the dye sensitizer to Ln ions, resulting in very high TET efficiencies; and (2) good device stability enabled by the fast depopulation of triplet states due to the efficient TET. This second point is notable, as material stability has been a limiting factor in applications of dye-sensitized LnNPs. Ultimately, the authors successfully measure external quantum efficiencies (EQEs) in their devices, which brings me to my main worry: these EQEs are very low. My concern is that in the long run, the main result will be seen as a novelty rather than a breakthrough demonstration, with such low efficiency and no obvious path forward. I also feel more discussion of the advantages of their device in the context of existing technologies is needed. For this reason, I believe publication in Nature would be premature.

Comments:

- 1) Figure 1d plots emission bandwidth (y-axis) and wavelength (x-axis) for their device in comparison to other SWIR LEDs. As the authors note, only their device has the narrow bandwidth in this spectral region. A main reason for this is that this is the regime of laser diodes: for narrow bandwidth, the field has chosen to go this direction rather than attempt to optimize LED bandwidth. So I'm not sure this is a compelling comparison, and this raises the primary question, which for some reason is not discussed in the manuscript: How is their device better than laser diodes in this wavelength range? What are the big advantages that LnLEDs bring to the table?
- 2) Currently the manuscript/SI is lacking a more thorough, quantitative comparison to the existing and previously demonstrated devices such as the QDLEDs, OLEDs along with more standard semiconductor LEDs. Readers would benefit from a table that contains a more comprehensive description of the other SWIR LEDs (and laser diodes), including emission wavelength, bandwidth, EQE, drive voltage and current, etc.
- 3) The authors note that the dye is a limiting factor to the EQE, which can be addressed in the future. It seems the LnNP QE is another limit. What is quantum yield of the LnNPs and the dye-sensitized LnNP components themselves? These should be measured and reported, as they set another limit on achievable device QE.
- 4) The authors should perform a systematic study of dye-sensitized LnNPs with varying passivating shell thicknesses. Is no shell really the best? I see the trade-off between maximizing TET with no shell (maximizing Dexter ET) vs. greatly reduced surface losses with thicker shells. Is there an optimum?
- 5) The reference list is rather light, considering there exists significant amount of literature on dye-sensitized Ln-doped NPs.

Some key, relevant, works seem to be missing from the reference list, including but not limited to:
Wang, et al., "Dye-sensitized lanthanide-doped upconversion nanoparticles," Chem. Soc. Rev., 2017,46, 4150.
Garfield, et al., "Enrichment of molecular antenna triplets amplifies upconverting nanoparticle emission," Nature Phot., 2018,12, 402.

In summary, these are good preliminary results likely suitable for another journal at this stage, with a follow-up work building on these results then more suitable for Nature if they can demonstrate some of the promise noted in the intro and conclusion of this manuscript.

Referee #2

(Remarks to the Author)

Re: Molecular triplets turn on insulating lanthanide-doped nanoparticles...
by Zhongzheng Yu & Junzhi Ye, et al.

Lanthanides are the foundation of many important optical technologies, they exhibit sharp emission spectra, and emit variously across the visible and NIR. Lanthanides are also expected to surpass traditional fluorescent efficiency limits in OLEDs. Thus, it is not surprising that lanthanides are one of the earliest technologies studied in OLEDs. But their application to OLEDs has traditionally been limited by their low efficiency.

Traditionally, the best lanthanide matrices are crystals like YAG, where the optical transitions of the lanthanide ion are protected from loss mechanisms. But as noted in this work, these insulating crystalline matrices cannot easily be employed in OLEDs. Instead, lanthanides have been previously used as metal centers within molecules, much like traditional molecular Pt- or Ir-based phosphors, except that energy is transferred entirely to the lanthanide ion where it drives the emissive transition. But in these molecules the lanthanide is surrounded by a few, relatively floppy, C and H bonds. The low efficiency of lanthanides in OLEDs to date is thought to be caused by this inferior environment.

This work seeks to address the fundamental limitation of lanthanides in OLEDs by employing lanthanide ions within inorganic nanoparticles. While this can potentially solve the low emission efficiency, it creates a challenge in communicating the energy from the charge transport materials in the OLED. The latter problem is solved here by capping the nanoparticles with anthracene-based ligands.

The authors demonstrate that their energy transfer scheme is relatively successful and efficient using transient absorption in Fig. 3. The OLED characterization, however, yields remarkably low efficiencies 'around 0.001%'. This result significantly undermines the claims in the rest of the manuscript because neither triplet energy transfer, nor an improved atomic environment, manifestly improves the efficiency of the OLEDs.

Some clues to the poor performance might be observed in the IVL data shown in Fig. 4 e & f. The Er and Yb devices show nonlinearities potentially due to hysteresis and possibly charge trapping.

Scientifically, it would seem necessary to understand the apparently dramatic difference between the performance of these materials under optical vs electrical excitation. For example, do the materials trap charge? To study a control, the authors build an OLED using only the anthracene derivative (leaving out the nanoparticles). The control OLED exhibits a significantly different and much cleaner IV. But the efficiency is poor (0.5%) relative to a typical OLED, and the authors argue that this largely explains the ~0.001% performance of the lanthanide-OLEDs in the NIR. But this control is not very clean. For example, what is PL yield of the anthracene derivative? Also, the anthracene derivative presumably emits only from the singlet. And in any case, how does this explain the enormous difference between visible and NIR efficiency?

Unfortunately, the impact of this work is limited by the poor efficiency. While the nanoparticle matrix seems like a promising approach, it is difficult to believe claims of a new and promising technology unless we can also understand the origins of the enormous losses.

Referee #3

(Remarks to the Author)

Yu and co-authors have developed a type of LnNP@9-ACA heterogeneous NIR-II LnLEDs that operate at low voltages, exhibiting good resistance to high voltages and current densities. The demonstrated configuration and results indicate potential for the design and fabrication of new types of LEDs, particularly when used in conjunction with LnNPs. However, I have a major concern regarding the discussion of the energy transfer mechanism and the basic background knowledge concerning LnNPs. My detailed major comments are:

1. The first paragraph of the Introduction contains significant logical flaws:
 - 1) NaF has scarcely been explored as a host for lanthanides.
 - 2) "LnNPs have high photo- and chemical stability in various environments and have sharp and tunable emission, especially in the second near-infrared range (NIR-II, 1000-1700 nm)." The significance of NIR-II in comparison to other wavelengths regarding sharp and tunable emissions requires clarification.
 - 3) "This has motivated research into their applications in fluorescence microscopy, deep-tissue theranostics, optogenetics, sensing, and optical communication." it is worth noting that the motivation described does not precisely align with the examples provided, and references primarily focus on Upconversion nanoparticles rather than NIR-II emissions.

2. In Figure 1b, the energy transfer mechanism lacks supporting evidence. There is a considerable mismatch between the energy levels of the 9-ACA triplet and the lanthanides at NIR-II. This prompts questions regarding the feasibility of a direct energy transfer to the NIR-II rather than to other higher excited states. If such direct transfer is plausible, what is the efficiency of the process? Additionally, how does the issue of oxygen quenching impact the triple state?

3. In Figure 1C, Ho emission suggests a deviation from a standard f-f transition. To enhance clarity, it is essential to provide evidence for the attribution of this emission.

4. The comparison of FWHM in Figure 1d is distracting, especially with the addition of the "criterion for optical communication." This explanation seems off-topic. It would be more effective to elaborate on why a narrow band is preferred for LEDs.

5. The statement "OA also binds to Na+" requires clarification, supported by an analysis of the FTIR data. Additionally, it would be beneficial to explain how the replacement ratio was calculated.

6. The energy transfer efficiency, ligand exchange rate, and Ln³⁺-Triplet interaction need to be analyzed cohesively to establish a comprehensive understanding for constructing an optimal hybrid system.

7. Detailed discussions should be added to clarify the mechanism for the ISC efficiency for 9-ACA attached to LnNPs. Why did different doping ions have different ISC efficiency? What is the relation between ISC efficiency and EQEs? Why they show different rules? The doping concentration mentioned 20 mol% on page 3 line 83 is not enough to clarify the mechanism. Experiments by tuning concentration also should be done in case to give an approach to tuning the performance of LnLEDs.

8. What about the stability of the LnLEDs?

9. Further discussions, like comparison with traditional LEDs, should be added to clarify the significance of a low turn-on voltage, a high voltage tolerance, and a current intensity.

Minor comments:

1. In Figure 1C and Figure 2e, it is imperative to accurately label all correlated transitions of the lanthanides to ensure a comprehensive and clear depiction of the data.

2. In Figure 2c and 2d, it is essential to specify which Ln nanoparticles correspond to the FTIR spectra. Furthermore, comprehensive FTIR curves for all other LnNPs, both before and after ligand exchange, should be included in the Supplementary Information

3. Figure S6 should feature the standard hexagonal phase XRD of nanoparticles.

Version 2:

Reviewer comments:

Referee #1

(Remarks to the Author)

I have read through the response letter, the new manuscript, and the new supplementary information file in detail. The authors are to be commended for the impressive suite of new experiments conducted since the previous submission, resulting in mechanistic elucidation and, ultimately, to significant device improvements including orders-of-magnitude increase in EQE. The low efficiency was my primary concern in the previous round, and the fact that they now present numbers that are on par with/better than existing systems is impressive – and at the same time addresses my worry that there was no reasonable path towards improvement. While I am not sure that one should expect innovations over time comparable to those shown by the perovskite system the authors mention, they do make a compelling case that future improvements can also be possible.

I feel the added comparisons to other existing systems make their arguments stronger (Suppl. Tables 1-3). I also appreciate the initial core-shell mechanistic studies and their discussion of the tradeoffs offered by shells, and further agree that this is a topic for future study and improvement. I therefore find the updated manuscript novel, interesting, and potentially impactful, and thus recommend it for publication.

Referee #2

(Remarks to the Author)

Re: Molecular triplets turn on insulating lanthanide-doped nanoparticles....., by Zhongzheng Yu, Zunzhou Deng, Junzhi Ye, et al.

This revised version has addressed the substantial weakness with the initial manuscript, notably the very low quantum efficiency. The core-shell Yb/Nd nanoparticles exhibit up to 0.6% (including an hemispherical outcoupling lens), which is

now respectable if still far from the best QD devices.

In my view, the primary merit of the manuscript remains the interesting energy transfer scheme, where triplets are conveyed to the surface of the nanoparticles, and then some presumably find their way into the bulk. This scheme apparently solves the traditional problem with sensitizing rare earths - their inability to emit efficiently when directly coupled to an organic sensitizer. I suspect that many readers may find it frustrating that energy transfer within the nanoparticles largely remains obscure. Consequently, I'd encourage the authors to present what data they have concerning the difference between PL and EL (especially if the pump wavelength is such as to compare spectral differences when exciting the bulk directly versus via the surface). There are clearly important differences between PL and EL e.g. Fig. S25, but these pass unnoted by the authors. Also, the efficiency improvement looks larger (100X?) than the PL difference (10X?). I think this merits a discussion of the potential importance of energy transfer away from the NP surface.

A related concern is that the core-shell work is treated almost as an appendage even though its introduction has resulted in a ~100X improvement. I would consider updating the entire manuscript, starting with the device and energy transfer schematics in Fig. 1. The existing Fig. 4 is not especially interesting. 4c-f could probably sit comfortably in the supplementary. On the other hand, Fig. S25 should probably be in the main text.

The authors have included a table of comparable devices. While EQE is an important comparison point, the other crucial device characteristic that must be highlighted is the very low brightness of the devices. I'd expect that the QDs can get to much higher brightness if their emission is faster than the emissive states in the rare earths. Indeed, Figs. 4g-i show that the rare earth emission barely reaches 1 mW/sr/m², perhaps because the excited states are saturated? The rapid roll-off in the EQE shown in Fig. S25 f suggests that the core-shell nanoparticles are also hitting a low ceiling (although this is not directly quantified). Fortunately, the narrow emission of the rare earths is compatible with optical cavities that might speed up emission and at least partially address this issue. It still needs to be noted, however.

A few other minor comments:

- lines in the figures are frequently so thin as to be barely legible
- Fig. 2e needs a legend
- line 268 of the supplementary should presumably refer to Fig. S23

Referee #3

(Remarks to the Author)

The authors have addressed some of my concerns; however, I still have concerns regarding the energy transfer study. A critical issue lies in the reported TET efficiencies, which are calculated to be 97.1%, 99.4%, and 98.5% for the Nd/Yb/ErNP@9-ACA nanohybrids. These values appear questionable given that the ligand exchange rate between 9-ACA and OA is reported to be only about 1%–6.8%, and that the emitters themselves are doped at only 20%. It seems unlikely that the small fraction of 9-ACA molecules bound to the surface could effectively interact with emitters and yield such high TET efficiencies. Upon closer examination, it appears that in the estimation of TET and throughout the spectroscopic studies, 9-ACA is consistently treated as the energy donor—perhaps contributing to the high TET efficiencies being reported. However, this assumption may not be entirely valid. Moreover, the core-shell design used for the LED device in the latter part of the work appears inconsistent with the main findings of the spectroscopic study. The doping strategy does not align with the conclusions drawn from the optical analysis, resulting in a disconnect between the fundamental photophysical insights and the final device architecture. My detailed comments are shown below:

1. In Fig. S5 and Fig. S6, why was the binding simulation conducted using Yb³⁺ or Y ions rather than Gd³⁺? Given that Gd³⁺ serves as the primary host ion in the nanoparticles, it would seem more appropriate to calculate the spectral shift based on Gd³⁺ coordination.
2. Why is the replacement ratio of Nd higher than that of Er, and significantly higher than that of Yb?
3. Why does the PL enhancement ratio in Figure 2 differ from the quantum efficiency enhancement ratio reported in Table S4? For instance, Nd-doped nanoparticles exhibit the highest quantum efficiency enhancement of 19.5-fold compared to Yb and Er. However, in the PL study, Nd shows the lowest enhancement, only 6.6-fold, which contradicts the trend observed in quantum efficiency.
4. The description of “PLQE measurements show that bound 9-ACA molecules on LnNPs have significantly decreased PLQE compared to pristine 9-ACA, consistent with the enhanced intersystem crossing (ISC) by coupling with LnNPs and efficient energy transfer to the Ln³⁺ ions (Supplementary Fig. 1)”, may be misleading. The reduced PLQE of bound 9-ACA is expected regardless of energy transfer, due to factors such as surface quenching or restricted molecular motion. Therefore, the decrease in PLQE cannot be unambiguously attributed to enhanced ISC or energy transfer to Ln³⁺ ions. In other words, the claim of ISC enhancement in the hybrid nanoparticles should not be based on a comparison with pristine 9-ACA alone. Instead, it should be evaluated relative to 9-ACA bound to free Ln³⁺ ions or host nanoparticle NaGdF₄ without doping (energy transfer). A similar concern applies to the interpretation of the lifetime data in Fig. 3b, where the observed shortening is used to conclude “quenching of the excitations and pointing to ISC enhancement and energy transfer.” This issue also extends to the pump-probe spectroscopy in Fig. 3d and the calculation of TET efficiency.
5. The core-shell nanoparticle design appears arbitrary and inconsistent with the conclusions drawn from the energy transfer efficiency analysis. In Supplementary Fig. S11, 50% Yb³⁺ doping outperforms 20%, with the explanation that “increasing the doping ratio of Yb³⁺ significantly enhances the downconversion intensities for both YbNPs and YbNP@9-ACA nanohybrids, as cross-relaxation between Yb³⁺ ions is not a major loss channel, unlike in Er³⁺ and Nd³⁺.” This suggests that high Nd³⁺ doping is unfavorable due to cross-relaxation losses. However, the core-shell design adopts the

opposite approach, using higher Nd³⁺ and low Yb³⁺ concentrations. This contradiction undermines the coherence between the spectroscopic findings and the structural design of the nanohybrids, making the optical study disconnected from the actual device architecture.

Version 3:

Reviewer comments:

Referee #1

(Remarks to the Author)

In the previous round of reviews, referees' raised valid points and questions regarding the mechanistic framework and related comparisons between different measurements, control studies, and improving clarity. In particular, I agree that an expanded comparison and presentation of PL and EL results, and the use of GdNP@9-ACA hybrids (and not just pristine 9-ACA) for quantitative comparison were both important suggestions. The comments also highlighted several descriptions in the manuscript that were susceptible to potential misunderstanding or ambiguity, and thus warranted further clarification. In my view, the responses and corresponding updates/changes in the manuscript and SI have nicely addressed these points (for example, see the updates to the new fig. 4 and fig. 3d), and have provided citations to relevant literature support where necessary. The work reveals several interesting directions that will benefit from further study by the larger community (and thus are outside the scope of the current report). I believe the exciting results (the (first-ever) demonstration of electrical pumping in these systems) and supporting evidence as presented here are suitable and worthy for publication.

Referee #2

(Remarks to the Author)

I think the revision overall looks good, except that I am still concerned about the brightness. The authors now state 'Future experimental and theoretical work is also required to enhance the brightness and operational stability of LnLEDs...'

I am not sure this comment is really sufficient, and it also doesn't look like the brightness comparison was added to the table in the supplementary.

Also there is no analysis of the brightness limitations. For example, the current density that we might expect to saturate the quantum dots is:

$J_{\text{sat}} = q N d / t$, where q is the electron charge, N is the density of emissive sites, d is the thickness of the emissive layer, and t is the lifetime of the emitter.

In this work $d=15\text{nm}$. Thicker layers (as might be used in conventional applications of rare earth phosphor coatings) are challenging to implement because of the insulating character of the material.

The key question is what is N ? In particular, can individual quantum dots support multiple excitations?

Is there any experimental evidence that these dots can support multiple excitations at a time without loss? If not, at a dot volume of 1000 nm^3 , $N=1\text{e}18\text{ cm}^{-3}$. Also assuming $t=1\text{ms}$ (I couldn't find transient data for Nd, Yb and Er dots in the manuscript).

These assumptions yield $J_{\text{sat}} = 0.2\text{mA/cm}^2$, which by eye looks very similar to the saturation that is experimentally observed in the devices Fig. 4i. These devices are behaving exactly as might be expected for materials that exhibit quenching for multiple excitations per quantum dot.

Referee #3

(Remarks to the Author)

The authors have addressed my question satisfactorily. I have no further concerns and therefore support the manuscript in its current form.

Manuscript Title: Molecular triplets turn on insulating lanthanide-doped nanoparticles for NIR-II light-emitting diodes

Referees' comments:

Referee #1 (Remarks to the Author):

This work describes the first demonstration of electrically-pumped emission from Ln-doped nanoparticles, overcoming an important barrier in these materials. They successfully fabricated the LED devices by combining dye-sensitized LnNPs with robust architecture concepts used in other device types. They show relatively narrow emission at various shortwave infrared (SWIR) frequencies, depending on the Ln species. Beyond the electrical excitation, there are some other excellent results here, including (1) a dye-NP attachment strategy that allows for direct bonding of the dye sensitizer to Ln ions, resulting in very high TET efficiencies; and (2) good device stability enabled by the fast depopulation of triplet states due to the efficient TET. This second point is notable, as material stability has been a limiting factor in applications of dye-sensitized LnNPs.

We thank the Referee #1 for the excellent summary and high praise of our work. We also thank the referee for the comments, which we have fully taken onboard and have significantly improved the manuscript.

Ultimately, the authors successfully measure external quantum efficiencies (EQEs) in their devices, which brings me to my main worry: these EQEs are very low. My concern is that in the long run, the main result will be seen as a novelty rather than a breakthrough demonstration, with such low efficiency and no obvious path forward. I also feel more discussion of the advantages of their device in the context of existing technologies is needed. For this reason, I believe publication in Nature would be premature.

To solve the Referee #1's main concern, we have now greatly improved the device performance and validated the paths towards how to further increase EQE in the future. Our strategies include: (1) optimizing the optical structure of the LnLEDs, which boost the light out-coupling efficiencies across the NIR-II range; (2) applying a new electron transport layer (ETL) material, TmPyPB, 1,3,5-Tris(3-pyridyl-3-phenyl)benzene, with one-order-of-magnitude higher electron mobility than that of the ETL in our previous devices; (3) performing appropriate encapsulation of the device to exclude interference from ambient oxygen or water; (4) utilizing core-shell LnNPs with significantly improved photoluminescence quantum efficiency (PLQE); (5) utilizing a new hole-transport layer (HTL) material to promote hole-injection and suppress electron overflow from the emitting layer. These strategies are based on our understanding of the efficiency-loss channels in our novel LnLEDs and have been validated by the progressively improved EQE. Our new optical structure is shown in Fig. R1 in comparison with the previous device.

The peak EQE of the LnLEDs has now been significantly improved to 0.64% (Fig. R2), which is higher than most SWIR OLEDs emitting at above 1000 nm (Y. Xie et al., Nat. Photonics 16, 752, 2022). The current peak EQEs of our LnLEDs have been improved by more than **640-fold** compared

to the first submission, which indicate that **our device-optimization approaches represent a rational and promising route toward better device performance.**

Fig. R1 a, The cross-sectional HAADF STEM images of the old structure and new structure to show the change of the thickness of different layers. The thickness of all the functional layers have been optimized to boost the NIR light out-coupling efficiency of the emissive layer. **b**, Simulated NIR light-extraction efficiencies in NIR regime. The light out-coupling efficiencies in the range of 900-1500 nm has been significantly improved in the new device structure. The new ETL (TmPyPB) offers a higher electron mobility (μ_e), which is important for applying the new optical structure with thicker charge-transport layers.

Fig. R2 The comparison of NIR EQE of the optimized LnLEDs and original LEDs in the first submission. The peak EQEs of the optimized have been significantly improved and there are still many paths to boost the EQEs of LnLEDs.

In the revised manuscript, we have updated the device structure and device performance with those from the new structure. We also highlight in the manuscript the importance of optical design on the light coupling in NIR-II spectral regime. To further enhance the NIR EQEs of LnLEDs, we utilized core-shell Yb@Nd LnNPs in the form of $\text{NaGd}_{0.8}\text{F}_4:\text{Yb}_{0.2}@\text{NaGd}_{0.4}\text{F}_4:\text{Nd}_{0.6}$ as emitting materials. We also propose other potential approaches to improve the device performance in the

future. For example, using charge-transport molecules with higher mobilities, organic sensitizer with higher quantum yields, or applying more sophisticated photonics structures to resonant light out-coupling of the emission peak.

We would like to note that the improvement of EQEs for a new-type device requires the effort of a broad community and takes some time and follow-up work to be achieved. For instance, the first report of perovskite LEDs from within our own group showed a relatively low EQE of 0.1% for green emission (Z. K. Tan et al., Nat. Nanotechnol. 9, 687, 2014). This has now been improved to 28.9% or even higher in less than a decade (J. S. Kim et al., Nature 611, 688, 2022). Further work by the community, following on from this first proof-of-concept work will no doubt greatly increase device efficiency.

Our key aim here is the first demonstration of insulating LnNP-based NIR-II LEDs, to provide a novel method to overcome the insulating nature of LnNPs and provide a new excitation source to utilize the sharp and stable emission from LnNPs in the NIR-II range. We believe this proof-of-concept work establishes that LnNPs can be electrically excited, an important breakthrough in the field, which has been a goal for LnNP research community for several decades. Currently LED technology in this wavelength region is limited to expensive III-V materials such as InGaAs. LEDs based on colloidal quantum dots such as PbS, HgTe etc are limited by their homogenous broadening to very large emission linewidths (> 150 nm) in this region, making them unsuitable for many optical communication or chemical/biomedical imaging/sensing applications. The narrow linewidths we achieve here (~20 nm), combined with the inherent ease of processing, flexibility, wide-area compatibility and potential low-cost of organic-LnNPs hybrids offers exciting possibilities for a new generation of light sources across the NIR-I and NIR-II ranges.

Comments:

1) Figure 1d plots emission bandwidth (y-axis) and wavelength (x-axis) for their device in comparison to other SWIR LEDs. As the authors note, only their device has the narrow bandwidth in this spectral region. A main reason for this is that this is the regime of laser diodes: for narrow bandwidth, the field has chosen to go this direction rather than attempt to optimize LED bandwidth. So I'm not sure this is a compelling comparison, and this raises the primary question, which for some reason is not discussed in the manuscript: How is their device better than laser diodes in this wavelength range? What are the big advantages that LnLEDs bring to the table?

We thank the Referee for raising this point. We note that laser diodes generate coherent light, which have totally different working mechanism and different application scenarios compared with non-coherent light sources like LEDs. Also, most existing NIR-II laser diodes and their driving electronics are much more expensive than LEDs. It is also difficult to have flexible NIR-II lasers based on III-V technologies, that currently dominate this space. Thus, NIR-II LEDs have the potential to offer low-cost, flexible light sources which could open up a range of applications in communication, chemical, biomedical imaging and sensing applications. We consider that the appropriate comparison is thus with different NIR-II LEDs (rather than lasers), especially for the first demonstration of a new type insulating LnNP-based LEDs.

2) Currently the manuscript/SI is lacking a more thorough, quantitative comparison to the existing and previously demonstrated devices such as the QDLEDs, OLEDs along with more standard semiconductor LEDs. Readers would benefit from a table that contains a more comprehensive description of the other SWIR LEDs (and laser diodes), including emission wavelength, bandwidth, EQE, drive voltage and current, etc.

We appreciate the suggestion from Referee #1 and have added a table containing a more comprehensive description of other SWIR LEDs in SI and shown as below (Supplementary Table 1) and with laser diodes (Supplementary Table 2).

Supplementary Table 1. Description of typical existing SWIR LEDs without additional light-outcoupling structures.

Emissive material	EL Peak (nm)	FWHM (nm)	Turn-On Voltage (V)	EQE (%)	Current Density at Peak EQE (mA cm ⁻²)	Year /ref
Nd-doped GaN	1082,1364	~50	~150	-	-	2004 ¹
Erbium tris(8-hydroxyquinoline)	1533	33	12	N/A	-	1999 ²
Rrbium organic compound	1540	~100	>10	N/A	<3	2000 ³
Conjugated polymer-Yb porphyrin blends	977	N/A	4	0.001	<200	2001 ⁴
Conjugation Platinum(II) Porphyrins	1005	>200	8.3	0.12	8-10	2011 ⁵
Selena-diazole containing polymers	990	>200	23.5	0.018	~20	2015 ⁶
D-p-A-p-D NIR chromophores	1050	220	4.8	0.33	0.1	2009 ⁷
D-A-D conjugated polymer	970	~150	1.1	0.05	<1	2004 ⁸
Single-walled carbon nanotubes	1177	50	2.1	0.01	<20	2018 ⁹
Y6	1040	~200	1.07	0.25	~10-100	2022 ¹⁰
Y11	1035		1.01	0.33		
IDSe-Th	1015		1.08	0.054		
Y-2W-F	1104		1.07	0.006		
SiOTIC-4F	1120		0.91	0.031		
COTIC-4F	1200		0.78	0.031		
TADF	1010	>250	3	0.003	20	2020 ¹¹
Yb(DBM) ₃ (H ₂ O) ₂ : DPE PO	1190	-	5-6	0.15	1	2017 ¹²

ErQ ₃ Singlet Fission OLED	1530	80	5	-	-	2018 ¹³
PbS in MEH-PPV	1160	190	5-6	0.27	-	2020 ¹⁴
InAs-ZnSe in MEH-PPV	1300	200	2.0	0.5	-	2002 ¹⁵
PbSe in MEH-PPV	1280	130	3.0	0.83	30	2003 ¹⁶
PbS QDs	1054	120	0.7-2.1	2.00	4	2012 ¹⁷
PbS-CdS QDs	1240	180	1.4	4.3	13.85	2015 ¹⁸
PbS in Perovskite	1391	150	<1	5.2	7.4	2016 ¹⁹
PbS QD-in-QD	1400	150	0.6	7.9	1-2	2019 ²⁰
PbS QD-in-QD	1400	150	0.7	8	200-400	2020 ²¹

References

- Kim, J. H. & Holloway, P. H. Near-infrared electroluminescence at room temperature from neodymium-doped gallium nitride thin films. *Applied Physics Letters* 85, 1689-1691, (2004).
- Curry, R. J. & Gillin, W. P. 1.54 μm electroluminescence from erbium (III) tris(8-hydroxyquinoline) (ErQ)-based organic light-emitting diodes. *Applied Physics Letters* 75, 1380-1382, (1999).
- Sun, R. G., Wang, Y. Z., Zheng, Q. B., Zhang, H. J. & Epstein, A. J. 1.54 μm infrared photoluminescence and electroluminescence from an erbium organic compound. *Journal of Applied Physics* 87, 7589-7591, (2000).
- Harrison, B. S. et al. Near-infrared electroluminescence from conjugated polymer/lanthanide porphyrin blends. *Applied Physics Letters* 79, 3770-3772, (2001).
- Graham, K. R. et al. Extended Conjugation Platinum(II) Porphyrins for use in Near-Infrared Emitting Organic Light Emitting Diodes. *Chemistry of Materials* 23, 5305-5312, (2011).
- Tregnago, G., Steckler, T. T., Fenwick, O., Andersson, M. R. & Cacialli, F. Thia- and seleno-diazole containing polymers for near-infrared light-emitting diodes. *Journal of Materials Chemistry C* 3, 2792-2797, (2015).
- Qian, G. et al. Simple and Efficient Near-Infrared Organic Chromophores for Light-Emitting Diodes with Single Electroluminescent Emission above 1000 nm. *Advanced Materials* 21, 111-116, (2009).
- Chen, M. et al. 1 micron wavelength photo- and electroluminescence from a conjugated polymer. *Applied Physics Letters* 84, 3570-3572, (2004).
- Graf, A., Murawski, C., Zakharko, Y., Zaumseil, J. & Gather, M. C. Infrared Organic Light-Emitting Diodes with Carbon Nanotube Emitters. *Advanced Materials* 30, 1706711, (2018).
- Xie, Y. et al. Bright short-wavelength infrared organic light-emitting devices. *Nature Photonics* 16, 752-761, (2022).
- Liang, Q., Xu, J., Xue, J. & Qiao, J. Near-infrared-II thermally activated delayed fluorescence organic light-emitting diodes. *Chemical Communications* 56, 8988-8991, (2020).
- Jinnai, K., Kabe, R. & Adachi, C. A near-infrared organic light-emitting diode based on an Yb(iii) complex synthesized by vacuum co-deposition. *Chemical Communications* 53, 5457-5460, (2017).
- Nagata, R., Nakanotani, H., Potscavage Jr, W. J. & Adachi, C. Exploiting Singlet Fission in Organic Light-Emitting Diodes. *Advanced Materials* 30, 1801484, (2018).
- Konstantatos, G., Huang, C., Levina, L., Lu, Z. & Sargent, E. H. Efficient Infrared Electroluminescent Devices Using Solution-Processed Colloidal Quantum Dots. *Advanced*

Functional Materials 15, 1865-1869, (2005).

15 Tessler, N., Medvedev, V., Kazes, M., Kan, S. & Banin, U. Efficient Near-Infrared Polymer Nanocrystal Light-Emitting Diodes. *Science* 295, 1506-1508, (2002).

16 Steckel, J. S., Coe-Sullivan, S., Bulović, V. & Bawendi, M. G. 1.3 μm to 1.55 μm Tunable Electroluminescence from PbSe Quantum Dots Embedded within an Organic Device. *Advanced Materials* 15, 1862-1866, (2003).

17 Sun, L. et al. Bright infrared quantum-dot light-emitting diodes through inter-dot spacing control. *Nature Nanotechnology* 7, 369-373, (2012).

18 Supran, G. J. et al. High-Performance Shortwave-Infrared Light-Emitting Devices Using Core-Shell (PbS-CdS) Colloidal Quantum Dots. *Advanced Materials* 27, 1437-1442, (2015).

19 Gong, X. et al. Highly efficient quantum dot near-infrared light-emitting diodes. *Nature Photonics* 10, 253-257, (2016).

20 Pradhan, S. et al. High-efficiency colloidal quantum dot infrared light-emitting diodes via engineering at the supra-nanocrystalline level. *Nature Nanotechnology* 14, 72-79, (2019).

21 Pradhan, S., Dalmases, M., Baspinar, A.-B. & Konstantatos, G. Highly Efficient, Bright, and Stable Colloidal Quantum Dot Short-Wave Infrared Light-Emitting Diodes. *Advanced Functional Materials* 30, 2004445, (2020).

Supplementary Table 2. Description of typical commercial SWIR laser diodes

Type	Emission Peak (nm)	FWHM (NM)	Operating Voltage (V)	Operating Current (mA)
L980H1	980 nm	0.5-2	2.0-2.5	300
L980P030	980 nm	10	1.5-2.0	50-70
LPS-1060-FC	1064 nm	10	1.4-2	300
LP1310-PAD2	1310	3	1-2	20-40
L1480G1	1480	20	1.6	5000-8000
LP1550-SAD2	1550	3	1-2	20-40
L1550G1	1550	20	1.5	5000-8000

All these diodes can be purchased on THORLABS, technical data can be found on the spec sheet online (https://www.thorlabs.com/newgrouppage9.cfm?objectgroup_id=4737). The prices of LP980-SA60 980 nm, L980H1, DBR1060PN 1060 nm, LPS-1060-FC 1064 nm, LP1310-PAD2 1310 nm, L1480G1 1480 nm, and LP1550-SAD2 1550 nm laser diodes are £520.61, £219.55, £4,171.54, £892.37, £595.68, £283.23, and £617.72, respectively.

3) The authors note that the dye is a limiting factor to the EQE, which can be addressed in the future. It seems the LnNP QE is another limit. What is quantum yield of the LnNPs and the dye-sensitized LnNP components themselves? These should be measured and reported, as they set another limit on achievable device QE.

We thank the Referee for the suggestion. This is indeed an important element to consider, and we agree that the PLQE of LnNP@9-ACA nanohybrid is another limit factor to EQE. The PLQEs of LnNPs and LnNP@9-ACA nanohybrids under the excitation of 375 nm have been measured and listed as below.

Supplementary Table 3. PLQEs of different LnNPs and LnNP@9-ACA nanohybrids.

Sample Name	NdNPs	NdNP@9-ACA
PLQE	0.02%	0.39%
Sample Name	YbNPs	YbNP@9-ACA
PLQE	0.02%	0.14%
Sample Name	ErNPs	ErNP@9-ACA
PLQE	0.01%	0.03%
Sample Name	Yb@NdNPs	Yb@NdNP@9-ACA
PLQE	0.59%	3.04%

Here in this first proof-of-concept demonstration, we use ultrasmall LnNPs without surface passivation and relatively high doping ratio (20%) of fluorescent Ln^{3+} ions in LnNPs. The single Ln^{3+} dopant without core-shell structure will allow us to understand the energy transfer pathways and efficiencies in this new NIR-II emissive hybrid system. The PLQEs of LnNPs and LnNP@9-ACA nanohybrids could definitely be further improved, by optimization of the structure of LnNPs, especially for tuning the doping ratio of Ln^{3+} and design of core-active shell structure.

To improve the PLQEs of LnNPs and thus boost the EQEs of the LnLEDs, we have fabricated core-shell Yb@Nd (in the form of $\text{NaGd}_{0.8}\text{F}_4:\text{Yb}_{0.2}@\text{NaGd}_{0.4}\text{F}_4:\text{Nd}_{0.6}$) LnNPs, with an average size of ~ 10 nm using the $\text{Yb}_{0.2}$ cores with an average size of ~ 6 nm (See Supplementary Fig.25). The PLQEs of Yb@NdNPs and Yb@NdNP@9-ACA hybrids have been significantly improved to 0.59% and 3.04%, respectively, under 375 nm laser excitation. The significantly improved PLQE of the core-shell Yb@Nd LnNPs result in a further boosting in the EQE of the LnLEDs to 0.64%. Thus, we agree with Referee #1 that the PLQEs of LnNPs set another limit for the EQEs of LnLEDs.

In a recent report, the core-shell particles can show PLQY as high as 50% for Er^{3+} emission at 1530 nm (Arteaga Cardona, F. et al. Preventing cation intermixing enables 50% quantum yield in sub-15 nm short-wave infrared-emitting rare-earth based core-shell nanocrystals. *Nat. Commun.* 14, 4462, (2023).). This shows the potential for improvement in device performance in the future. However, as our LnLEDs are TET mediated devices, we must balance triplet injection into LnNPs and ligand coverage with issues like shell thickness. This will require in-depth future studies, which we consider beyond the scope of this first report. But core-shell structures with inherently higher PLQEs have been proven to be a clear and effective pathway forward to enhance the device performance in our work.

4) The authors should perform a systematic study of dye-sensitized LnNPs with varying passivating shell thicknesses. Is no shell really the best? I see the trade-off between maximizing TET with no shell (maximizing Dexter ET) vs. greatly reduced surface losses with thicker shells. Is there an optimum?

This is a very good question and we agree with the referee that there will be an optimum value to shell thickness, with thick shells blocking triplet transfer and very thin shells providing no boost in PLQE. The reason why we chose core only systems in this manuscript is to investigate and shed

light on the underlying energy transfer pathways and efficiencies between 9-ACA and LnNPs, including FRET and TET. We consider that such systems already present complicated dynamics and since this is the first report of such devices, we feel that the proof-of-concept demonstration would not be benefited by an in-depth structure-function study of shell thickness. Such an in-depth structure-function study of shell thickness is defiantly called for in future studies, and as we have remarked above, core-shell structures with inherently higher PLQE provide clear pathway forward to increase device performance.

We intended to systematically study the optimal passivating shell thickness and how best to optimize TET vs shell thickness in future studies. We will also study the design of active shell structure, where the shell contains an acceptor Ln^{3+} ion that receives triplets from the organic ligand on the surface and passes it on to a different Ln ion in the core, which is the emitter. This should allow for a much thicker shell, as there is a “energy conduction pathway” even within the shell and triplets do not have to tunnel across the entire shell thickness. However, both these areas require detailed future study and would benefit from input from the wider LnNP and LED communities. This is why we consider them beyond the scope of the current manuscript.

However, to give a preliminary indication of the effect of inert shells, we have carried out experiments with inert shells. We have coated an inert layer (NaGdF_4) on the surface of HoNP cores to form a core/shell structure ($\text{NaGd}_{0.8}\text{F}_4:\text{Ho}_{0.2}@\text{NaGdF}_4$, labelled as Ho@GdNPs) to study the influence of shell on TET efficiencies (see the TEM images in Fig. R1). We found out that the average size of HoNP core measured in TEM image is around 8.5 nm and average size of $\text{Ho}_{0.2}@\text{Gd}$ core@thin shell NPs is around 11 nm. The thickness of the thin shell is around 1.3 nm. Coating of the thin inert shell did not increase the PL intensity at the same weight concentration (20 mg ml^{-1}). After ligand exchange, $\text{Ho}_{0.2}@\text{GdNP}@9\text{-ACA}$ sample showed a 6.6-fold enhancement, while HoNP@9-ACA sample showed 12.1-fold enhancement, indicating that even a thin inert shell would partially block the triplet energy transfer. To obtain a thicker inert shell, the average size of $\text{Ho}_{0.2}$ core measured in the TEM image is around 9 nm and the average size of $\text{Ho}_{0.2}@\text{Gd}$ core@thick shell NPs is around 15 nm. Coating a thick shell of 3 nm will cause a 3.7-fold enhancement of the PL intensity, while the 3 nm shell will totally block the triplet energy transfer (Fig. R1f).

Fig. R3 TEM images of **a**, core HoNPs and **b**, Ho@GdNPs with thin inert shell with an average shell thickness of 1.3 nm. **c**, The comparison of NIR PL spectra of HoNPs, Ho@Gd core@thin shell NPs and Ho@Gd@9-ACA under the same measurement conditions under the excitation of 350 nm with the concentration of 20 mg ml⁻¹. TEM images of **d**, core HoNPs and **e**, Ho@GdNPs with thick inert shell with an average shell thickness of 3 nm. **f**, The comparison of NIR PL spectra of HoNPs, Ho@Gd core@thick shell NPs and Ho@Gd@9-ACA under the same measurement conditions under the excitation of 350 nm with the concentration of 20 mg mL⁻¹.

5) The reference list is rather light, considering there exists significant amount of literature on dye-sensitized Ln-doped NPs. Some key, relevant, works seem to be missing from the reference list, including but not limited to:

Wang, et al., "Dye-sensitized lanthanide-doped upconversion nanoparticles," *Chem. Soc. Rev.*, 2017,46, 4150.

Garfield, et al., "Enrichment of molecular antenna triplets amplifies upconverting nanoparticle emission," *Nature Phot.*, 2018,12, 402.

We have cited the references suggested by Referee #1.

In summary, these are good preliminary results likely suitable for another journal at this stage, with a follow-up work building on these results then more suitable for Nature if they can demonstrate some of the promise noted in the intro and conclusion of this manuscript.

We thank again Referee #1 for the raising these concerns and suggestions. We believe that the manuscript is significantly improved by the changes following the Referee #1's suggestions. We highlight that we now provide new type hybrid NIR-II LnLEDs with higher peak EQEs than most SWIR OLEDs emitting at above 1000 nm, overcome the insulating nature of LnNPs, and use low-voltage electricity as a new excitation source for LnNPs, which hopefully meet the high standard to

be published in Nature.

Referee #2 (Remarks to the Author):

Re: Molecular triplets turn on insulating lanthanide-doped nanoparticles...
by Zhongzheng Yu & Junzhi Ye, et al.

Lanthanides are the foundation of many important optical technologies, they exhibit sharp emission spectra, and emit variously across the visible and NIR. Lanthanides are also expected to surpass traditional fluorescent efficiency limits in OLEDs. Thus, it is not surprising that lanthanides are one of the earliest technologies studied in OLEDs. But their application to OLEDs has traditionally been limited by their low efficiency.

Traditionally, the best lanthanide matrices are crystals like YAG, where the optical transitions of the lanthanide ion are protected from loss mechanisms. But as noted in this work, these insulating crystalline matrices cannot easily be employed in OLEDs. Instead, lanthanides have been previously used as metal centers within molecules, much like traditional molecular Pt- or Ir-based phosphors, except that energy is transferred entirely to the lanthanide ion where it drives the emissive transition. But in these molecules the lanthanide is surrounded by a few, relatively floppy, C and H bonds. The low efficiency of lanthanides in OLEDs to date is thought to be caused by this inferior environment.

This work seeks to address the fundamental limitation of lanthanides in OLEDs by employing lanthanide ions within inorganic nanoparticles. While this can potentially solve the low emission efficiency, it creates a challenge in communicating the energy from the charge transport materials in the OLED. The latter problem is solved here by capping the nanoparticles with anthracene-based ligands.

We thank the Referee #2 for the overall positive assessment and providing a deep insight of the field.

The authors demonstrate that their energy transfer scheme is relatively successful and efficient using transient absorption in Fig. 3. The OLED characterization, however, yields remarkably low efficiencies 'around 0.001%'. This result significantly undermines the claims in the rest of the manuscript because neither triplet energy transfer, nor an improved atomic environment, manifestly improves the efficiency of the OLEDs.

Some clues to the poor performance might be observed in the IVL data shown in Fig. 4 e & f. The Er and Yb devices show nonlinearities potentially due to hysteresis and possibly charge trapping.

Scientifically, it would seem necessary to understand the apparently dramatic difference between the performance of these materials under optical vs electrical excitation. For example, do the materials trap charge? To study a control, the authors build an OLED using only the anthracene derivative (leaving out the nanoparticles). The control OLED exhibits a significantly different and much cleaner IV. But the efficiency is poor (0.5%) relative to a typical OLED, and the authors argue that this largely explains the ~0.001% performance of the lanthanide-OLEDs in the NIR. But this

control is not very clean. For example, what is PL yield of the anthracene derivative? Also, the anthracene derivative presumably emits only from the singlet. And in any case, how does this explain the enormous difference between visible and NIR efficiency?

Unfortunately, the impact of this work is limited by the poor efficiency. While the nanoparticle matrix seems like a promising approach, it is difficult to believe claims of a new and promising technology unless we can also understand the origins of the enormous losses.

We thank the referee for the comment and agree that this is a key question. To resolve the issues of poor efficiency, which is also the main concern of Referee #1, we have now improved the device performance and validated the path towards how to further increase the EQEs. Our strategies include: (1) optimizing the optical structure of the LnLEDs, which boost the light out-coupling efficiencies across the NIR-II range; (2) applying a new electron transport layer (ETL) material, TmPyPB, 1,3,5-Tris(3-pyridyl-3-phenyl)benzene, with one-order-of-magnitude higher electron mobility than that of the ETL in our previous devices; (3) performing appropriate encapsulation of the device to exclude interference from ambient oxygen or water; (4) utilizing core-shell LnNPs with significantly improved photoluminescence quantum efficiency (PLQE); (5) utilizing a new hole-transport layer (HTL) material to promote hole-injection and suppress electron overflow from the emitting layer. These strategies are based on our understanding of the efficiency-loss channels in our novel LnLEDs and have been validated by the progressively improved EQE (discussed in detail below). Our new optical structure is shown in Fig. R1 in comparison with the previous device.

The peak EQEs of the LnLEDs have now been significantly improved to 0.64% (Fig. R2), which is higher than most SWIR OLEDs emitting at above 1000 nm (Y. Xie et al., Nat. Photonics 16, 752, 2022). The current peak EQEs of our LnLEDs have been improved by more than **640-fold** compared to the first submission, which indicate that **our device-optimization approaches represent a rational and promising route toward better device performance.** Resulting from the improved device design, our new devices show cleaner JV curves up to over 15 V. The transitions from leakage regime (0-4 V) to injection-limited regime (around 5 V), and space-charge-limited regime (over 10 V) can be seen from the JV curves (Fig. 4), which is similar to typical shapes of solution-processed LEDs.

Fig. R1 a, The cross-sectional HAADF STEM images of the old structure and new structure to show

the change of the thickness of different layers. The thickness of all the functional layers have been optimized to boost the NIR light out-coupling efficiency of the emissive layer. **b**, Simulated NIR light-extraction efficiencies in NIR regime. The light out-coupling efficiencies in the range of 900-1500 nm has been significantly improved in the new device structure. The new ETL (TmPyPB) offers a higher electron mobility (μ_e), which is important for applying the new optical structure with thicker charge-transport layers.

Fig. R2 The comparison of NIR EQE of the optimized LnLEDs and original LEDs in the first submission. The peak EQEs of the optimized have been significantly improved and there are still many paths to boost the EQEs of LnLEDs.

For the control device by merely 9-ACA molecules, we use the same solution-processed method without host matrix to fabricate the OLED as we did for LnLEDs, which is not the commonly used vapor deposition process to obtain a proper thickness. The PLQY of 9-ACA is measured to be 34.5% in hexane/THF (v/v=9:1) solution. Thus, the EQE is not high. We have revised our claim in the manuscript to avoid confusion.

One of the most obvious losses would be the low PLQY of ultrasmall core-only (<10 nm) and highly doped LnNPs (see Supplementary Table 3 above), which is around 0.02%. The PLQE of LnNP@9-ACA is improved, while still below 1%. Even if all the triplet excitons could be transferred to LnNPs, the EQE of LnNPs would not be high. In the revision, we have applied fabricated core-shell Yb@Nd (in the form of NaGd_{0.8}F₄:Yb_{0.2}@NaGd_{0.4}F₄:Nd_{0.6}) LnNPs, with an average size of ~10 nm using the Yb_{0.2} cores with an average size of ~6 nm. The PLQE of Yb@NdNP@9-ACA hybrids has been significantly improved 3.04% under 375 nm laser excitation. The significantly improved PLQE of the core-shell Yb@Nd LnNPs result in a further boosting in the EQE of the Yb@Nd LEDs to 0.64%. We thus validate this significant loss channel, and successfully suppress this loss channel to yield a significantly improved EQE. This loss channel could be further suppressed in future studies, by for instance changing the particle size, tuning the doping ratio, design of core-shell structure and multi Ln³⁺. For instance, a recent report has shown PLQY to 50% for Er³⁺ emission at 1530 nm in core-shell particles (Arteaga Cardona, F. et al. Preventing cation intermixing enables 50% quantum yield in sub-15 nm short-wave infrared-emitting rare-earth based core-shell nanocrystals. Nat. Commun. 14, 4462, (2023).).

Another notable loss is the electron-hole recombination in HTL or at HTL/ETL interface. This can be identified by the visible EL emission in all the LnLEDs shown in Fig. R4. The EL at ~ 610 nm resembles that of the exciplex formed at the interface of poly-TPD and organic ETL (Yin, Y. et al., *Journal of Materials Science: Materials in Electronics* 28, 19148-19154, (2017)). This feature indicates the electron-hole recombination at the direct contact of HTL and ETL, in voids of our monolayer LnNP@9-ACA nanohybrids, without forming excitons on the hybrids. The EL at ~ 430 nm corresponds to the spectra of poly-TPD, indicating that excitons are formed inside the HTL due to electron overflow from the emitting layer.

Fig. R4 The visible EL emission spectra of **a**, NdLED; **b**, YbLED; and **c**, ErLED at different voltages.

The electron leakage can be mitigated by using HTL with better hole-injection and electron-blocking properties. In this revision, we validated this strategy by using a new HTL, better hole-injection and electron-blocking capability (Bian, Y. et al. *Nature* 635, 854–859 (2024)). As shown by Fig. R5, PF8Cz has a HOMO level of -5.4 eV, 0.2 eV deeper than that of poly-TPD. The reduced hole-injection barrier from HTL to 9-ACA shall promote the exciton formation on 9-ACA before electron leakage from 9-ACA to the HTL. In addition to deeper HOMO, PF8Cz has been shown to suppress electron leakage from the emissive layer due to a high LUMO level and lower tail-state distributions in the band gap (Deng, Y. et al. *Nat. Photon.* 16, 505–511 (2022)). As shown by Fig. R5, on top of the strategy of core-shell configuration, the use of PF8Cz further enhanced the peak EQE of our LnLED from $\sim 0.1\%$ (yellow triangles) to $\sim 0.3\%$ (orange squares).

Fig. R5 Device optimization by using core-shell LnNPs, new HTL, and light-extraction structure. **a**, Energy-level diagram showing deeper HOMO of PF8Cz comparing to poly-TPD. **b**, Comparison of EQEs of the LnLEDs applying different strategies to suppress the efficiency-loss channels.

Another loss is lower light-extraction efficiency in NIR region. This efficiency loss channel is evidenced by a significant improvement of EQEs of all our LnLEDs by using the new optical structure

with adjusted film thickness (Fig. R1 and Fig. R2). While the light out-coupling efficiencies of our planar LnLED is still lower than visible LEDs (typically >25% for OLED), it is worth noting that there are plenty of engineering approaches to boost the light extraction. In revision, we have validated one of the widely used external optical structures by attaching a half-ball lens onto the substrate of the LnLED. The half-ball lens out-couples waveguided modes in the glass substrates, boosting the peak EQE from $\sim 0.3\%$ to $>0.6\%$.

Beside the strategies demonstrated, we also propose other potential approaches to improve the device performance in the future. For example, using charge-transport molecules with higher mobilities, organic sensitizer with higher quantum yields, or applying more sophisticated photonic structures to resonant light out-coupling of the emission peak. The main aim of this manuscript is to provide a new method to overcome the insulating nature of LnNPs and a new excitation method to activate LnNPs using relatively low voltages. We hope that the revisions made in this round have solved the main concerns of Referee #2 and significantly improved the quality of our manuscript.

Referee #3 (Remarks to the Author):

Yu and co-authors have developed a type of LnNP@9-ACA heterogeneous NIR-II LnLEDs that operate at low voltages, exhibiting good resistance to high voltages and current densities. The demonstrated configuration and results indicate potential for the design and fabrication of new types of LEDs, particularly when used in conjunction with LnNPs.

We thank the Referee #3 for the useful summary and positive assessment of our work.

However, I have a major concern regarding the discussion of the energy transfer mechanism and the basic background knowledge concerning LnNPs. My detailed major comments are:

1. The first paragraph of the Introduction contains significant logical flaws:

1) NaF has scarcely been explored as a host for lanthanides.

We apologize for the confusion on this, we have revised NaF to NaGd/Y/LuF₄.

2) "LnNPs have high photo- and chemical stability in various environments and have sharp and tunable emission, especially in the second near-infrared range (NIR-II, 1000-1700 nm)." The significance of NIR-II in comparison to other wavelengths regarding sharp and tunable emissions requires clarification.

We revised this sentence to "*LnNPs have high photo- and chemical stability in various environments and have narrow bandwidth and tunable emission in the second near-infrared range (NIR-II, 1000-1700 nm). This is in contrast to semiconductor-based systems, such as NIR-II emissive organic dyes or semiconducting colloidal quantum dots, which show broad emission spectra in this region.*"

3) "This has motivated research into their applications in fluorescence microscopy, deep-tissue theranostics, optogenetics, sensing, and optical communication." it is worth noting that the motivation described does not precisely align with the examples provided, and references primarily focus on Upconversion nanoparticles rather than NIR-II emissions.

Thanks for the kind suggestions, we have revised the statement and updated the references related to NIR-II emissions as listed below.

"This has motivated research into their applications in stimulated-emission depletion microscopy⁷, deep-tissue theranostics^{4,8-10}, sensing¹¹, and optical communication¹²."

⁴ Fan, Y. et al. Lifetime-engineered NIR-II nanoparticles unlock multiplexed in vivo imaging. *Nat. Nanotechnol.* 13, 941-946, (2018).

⁷ Liang, L. et al. Continuous-wave near-infrared stimulated-emission depletion microscopy using downshifting lanthanide nanoparticles. *Nat. Nanotechnol.* 16, 975-980, (2021).

⁸ Yu, Z., Chan, W. K. & Tan, T. T. Y. Neodymium-Sensitized Nanoconstructs for Near-Infrared Enabled Photomedicine. *Small* 16, 1905265, (2020).

⁹ Fang, Z. et al. Oxyhaemoglobin saturation NIR-IIb imaging for assessing cancer metabolism and

predicting the response to immunotherapy. Nat. Nanotechnol., (2023).

¹⁰ Pei, P. et al. X-ray-activated persistent luminescence nanomaterials for NIR-II imaging. *Nat. Nanotechnol.* 16, 1011-1018, (2021).

¹¹ Liu, L. et al. Er(3+) Sensitized 1530 nm to 1180 nm Second Near-Infrared Window Upconversion Nanocrystals for In Vivo Biosensing. *Angew. Chem. Int. Ed.* 57, 7518-7522, (2018).

¹² Liu, Y. et al. A photonic integrated circuit-based erbium-doped amplifier. *Science* 376, 1309-1313, (2022).

2. In Figure 1b, the energy transfer mechanism lacks supporting evidence. There is a considerable mismatch between the energy levels of the 9-ACA triplet and the lanthanides at NIR-II. This prompts questions regarding the feasibility of a direct energy transfer to the NIR-II rather than to other higher excited states. If such direct transfer is plausible, what is the efficiency of the process? Additionally, how does the issue of oxygen quenching impact the triple state?

We thank the Refree#3 for raising a very good question. The triplet energy of 9-ACA involves a wide range from 650 to 950 nm (1.3 to 1.9 eV). Thus, we do believe the triplet energy of 9-ACA is transferred to relatively high energy levels of Ln³⁺ and then relaxes to lower energy levels of Ln³⁺ to generate NIR-II emission, for example ⁴I_{13/2} of Er³⁺ to generate 1530 nm emission (⁴I_{13/2} → ⁴I_{15/2}). Considering the mismatch between the energy levels of 9-ACA and some low energy levels of Ln³⁺ (e.g. ⁴I_{13/2} for Er³⁺), it is possible that the direct energy transfer rate to these levels will be low and that instead, as suggested by the referee higher lying states play a major role in the TET. We have revised the Fig. 1b as below to avoid the confusion. However, it is challenging to distinguish which excited energy levels of Ln³⁺ the triplet transfer occurs to, since there are no optical signatures of these levels that can be probed easily via pump-probe spectroscopy (as used here to study the TET).

Fig. 1b, Simplified schematic showing electron and hole injection via organic molecules to turn on lanthanide ions in an insulating host lattice.

We measured the NIR PL intensities of YbNP@9-ACA nanohybrids under the O₂-free and exposure to air conditions as shown in Fig. R5. The NIR PL intensity from Yb³⁺ has been significantly quenched (53.3%) as O₂ quenches the triplet state of 9-ACA. This highlights the role of triplets in the energy

transfer process.

Fig. R5 NIR PL spectra of YbNP@9-ACA nanohybrids (concentration of 20 mg ml⁻¹) under 350 nm excitation show significant quenching in emission intensity of Yb³⁺ after exposure to air.

3. In Figure 1C, Ho emission suggests a deviation from a standard f-f transition. To enhance clarity, it is essential to provide evidence for the attribution of this emission.

Thank you for the pointing out that the EL of HoLED suggests a deviation from a standard f-f transition. We attribute the EL emission centered at around 910 nm to the broadened 910 nm emission of HoNPs ($^5I_5 \rightarrow ^5I_8$), consistent with previous report (Guhur, A. & Jackson, S. D. Efficient holmium-doped fluoride fiber laser emitting 2.1 μm and blue upconversion fluorescence upon excitation at 2 μm . *Optics Express* 18, 20164-20169, (2010).). But this peak is off the target of NIR-II EL, so we have removed all the data about HoNPs in the manuscript to enhance clarity and focus on Nd/Yb/Er-LEDs.

4. The comparison of FWHM in Figure 1d is distracting, especially with the addition of the "criterion for optical communication." This explanation seems off-topic. It would be more effective to elaborate on why a narrow band is preferred for LEDs.

Thanks for the suggestion. We have revised the claim and added more discussions about the importance of narrow band for LEDs in the manuscript.

5. The statement "OA also binds to Na⁺" requires clarification, supported by an analysis of the FTIR data. Additionally, it would be beneficial to explain how the replacement ratio was calculated.

We appreciate the suggestion and now have added the more discussions about the analysis of binding between OA and Na⁺. We now modify the main text to say:

"Density-functional theory (DFT) simulations (see Fig. 2 c-d and Supplementary Fig. 5-8 for details), indicate that the 9-ACA preferentially binds to the Ln³⁺ ion site on the surface of the LnNPs, in contrast to the oleic acid which also binds to the Na⁺ sites. DFT predicted FTIR spectra of 9-ACA

bonded to Gd^{3+} reproduces the experimentally observed spectrum, while 9-ACA bonded to Na^+ does not, and introduces peaks at 1600 cm^{-1} which are not observed (vertical lines in Fig. 2c). However, DFT predicted FTIR spectra for OA shows peaks shared at 1450 and 1590 cm^{-1} for OA bonded to Na^+ or Gd^{3+} (vertical lines in Fig. 2d)."

To determine the ligand replacement ratios, we identify unique FTIR signatures associated with the OA ligand and also the 9-ACA ligand on the nanocrystals in the solid state.

Determining the replacement ratio is first done by normalizing the OA-conjugated and 9-ACA-replaced device lanthanide spectra to the OA C-C stretch mode at $\sim 1595\text{ cm}^{-1}$ in Fig. R6. We then subtract both spectra to yield the red curve corresponding to the signatures due to ligand replacement of OA by 9-ACA. The blue shaded area corresponds directly to the $\sim 1595\text{ cm}^{-1}$ C-C stretch mode of the OA ligand. The red shaded region corresponding to the main 9-ACA C=C backbone stretch coordinated to a ligand, which can be confirmed by comparison to the fully replaced 9-ACA ligand spectrum above it in Fig. R6.

We then normalize for the different oscillator strength of these ligands by comparison to the DFT predicted spectrum, to give a normalized ratio between the OA and 9-ACA oscillator strength of $f_{ACA} = 1.48$. The area under each shaded region (A_{ACA} in red and A_{OA} in blue) may then be divided to obtain the replacement ratio R , with exact values listed in Table R1.

Fig. R6 Ligand replacement from FTIR spectra for Ho nanocrystals. Spectra of pure ACA ligand on lanthanide (9-ACA), of device with ACA replacing OA (device), of pure OA on lanthanide (OA), and the subtracted spectrum between device and OA (device-OA).

Table. R1 Ligand FTIR areas and replacement ratios

Ln^{3+}	9-ACA FTIR area (A_{ACA})	OA FTIR area (A_{OA})	Replacement ratio (R)
Er	0.70	12.6	3.6%
Nd	0.58	5.43	6.8%
Yb	0.16	10.2	1.0%

6. The energy transfer efficiency, ligand exchange rate, and Ln^{3+} -Triplet interaction need to be analyzed cohesively to establish a comprehensive understanding for constructing an optimal hybrid system.

We agree with the referee that these are all important points. Ligand exchange is a dynamic process and can be influenced by many factors and we note that ligands may detach from the surface during film formation, thus it is very difficult to have a quantitative value for the ligand density on the surface of the LnNP within the films.

To have better understanding of energy transfer efficiency, ligand exchange rate and Ln³⁺-triplet interaction in solution, we have performed additional experiments by changing the ligand exchange time of 9-ACA and studied the NIR emission from Yb³⁺ at different time points.

We find that the ligand exchange ratio is first increased by prolonging the reaction time and then reaches a plateau after around 80 min (Fig. R7a and b), by monitoring the absorbance change of YbNP@9-ACA nano hybrids. The nano hybrid system will inevitably involve free 9-ACA molecules even after washing, we thus have measured the photoluminescence excitation (PLE) spectra of YbNP@9-ACA nano hybrids (Fig. R7c) to show the amount of bound dye onto the surface of YbNPs. The PLE and NIR emission spectra show a similar trend with the absorption change (Fig. R7d-f). Thus, we can use the following ratio of peak emission of Yb³⁺/absorption of YbNP@9-ACA to evaluate the overall energy transfer efficiency to have a general idea of the energy transfer efficiency (Fig. R8). We find that the energy transfer efficiency is not significantly influenced by the ligand exchange ratio when the ligand exchange process reaches an equilibrium. Considering the near unity energy transfer efficiencies from the triplet states of 9-ACA to Nd/Yb/Er/HoNPs, the optimal hybrid system is mainly decided by tuning the amount of bound dye on the surface of LnNPs.

Fig. R7 (a) Absorption spectra of YbNP@9-ACA nano hybrids collected at different ligand exchange time and (b) corresponding peak absorption change with different ligand exchange time. (c) PLE spectra of YbNP@9-ACA nano hybrids collected at different ligand exchange time and (d) corresponding the corresponding peak intensities at 364 nm. (e) PL spectra of YbNP@9-ACA nano hybrids measured at different ligand exchange time and (f) corresponding peak emission intensity change with different ligand exchange time.

Fig. R8 The ratio of peak emission intensity of YbNP@9-ACA nanohybrids to the absorption of YbNP@9-ACA nanohybrids versus ligand exchange time.

7. Detailed discussions should be added to clarify the mechanism for the ISC efficiency for 9-ACA attached to LnNPs. Why did different doping ions have different ISC efficiency? What is the relation between ISC efficiency and EQEs? Why they show different rules? The doping concentration mentioned 20 mol% on page 3 line 83 is not enough to clarify the mechanism. Experiments by tuning concentration also should be done in case to give an approach to tuning the performance of LnLEDs.

The ISC efficiencies of organic molecules are influenced by the unpaired 4f electrons of different Ln³⁺ ions, which has been studied and reported in our previous publication (Han, S. et al. Lanthanide-doped inorganic nanoparticles turn molecular triplet excitons bright. *Nature* 587, 594-599, 2020). The ISC efficiencies are enhanced by the increased number of unpaired 4f electrons, which well explains why the HoNPs (4 unpaired 4f electrons) yield highest ISC efficiency and YbNPs (1 unpaired 4f electrons) yield the lowest ISC efficiency. However, the yields of triplets of 9-ACA under light and electrical excitations are different, with electrical excitation within the LEDs leading for 75% triplet formation as dictated by spin-statistics. Hence, we do not think it would be advisable to make comparisons between the ISC efficiency and EQEs.

The emission intensity of LnNPs is sensitive to doping concentrations for Ln³⁺ ions with ladder like energy levels due to the enhanced cross relaxation at higher doping concentrations. Tuning the doping concentration is one of the most common and straightforward methods to manipulate the NIR downconversion performance of LnNPs. To solve the concern of Referee #3, we have chosen to tune the doping concentration of Yb³⁺ which has only one energy levels and is not sensitive to cross relaxation at high doping concentration. We synthesized Yb_xNPs tuning different doping concentrations, including Yb5%, Yb20% and Yb50% and measured their NIR PL spectra (Fig. R9a). The NIR PL spectra is measured for different Yb_xNP@9-ACA (x=5%, 20% and 50%) nanohybrids under same conditions with a as shown in Fig. R7b. We found that the peak PL intensity from Yb³⁺ increased 2.1- and 6.1-fold, when increasing the doping ratio of Yb³⁺ from 5% to 20% and 50%. After ligand exchange, Yb_{0.05}@9-ACA showed 28.5-fold enhancement, Yb_{0.2}@9-ACA showed 34.1-

fold enhancement, Yb_{0.5}@9-ACA showed 54.1-fold enhancement (Fig. R9b). As Referee #3 suggested, these results proved that tuning the doping concentration of Ln³⁺ would be an effective approach to tune the performance of LnLEDs.

Fig. R9 NIR PL spectra of (a) YbNPs with different doping concentrations and (b) corresponding YbNP@9-ACA nanohybrids (concentration of 20 mg ml⁻¹) under 350 nm excitation.

8. What about the stability of the LnLEDs?

Our encapsulated devices ensure a consistent device performance during storage for over a month. We also characterized the operational stability of Yb@Nd LED by measuring the radiance decay upon constant current density. As shown by Fig. R10, the Yb@Nd LED demonstrate a half-life over 15 min. We believe this represents a reasonable stability given that our LnLED are the first proof-of-concept devices. We do note that device stability can be greatly improved over time for devices, for instance the first polymer OLEDs developed by our group showed half-life of 15 s and now commercial polymer-OLEDs approach lifetimes > 500k hours. Similarly, the first perovskite LEDs developed in our group (Z. K. Tan et al., Nat. Nanotechnol. 9, 687, 2014) showed lifetimes of around 300 s, which has now been greatly improved.

Fig. R10 Operational stability of Yb@Nd LED upon constant current density.

9. Further discussions, like comparison with traditional LEDs, should be added to clarify the significance of a low turn-on voltage, a high voltage tolerance, and a current intensity.

We appreciate the suggestion from Referee #3 and have added a table containing a more comprehensive comparison of other SWIR LEDs in SI and shown as below (Supplementary Table 1) along with further discussions in the manuscript.

Supplementary Table 1. Description of typical existing SWIR LEDs without additional light-outcoupling structures.

Emissive material	EL Peak (nm)	FWHM (nm)	Turn-On Voltage (V)	EQE (%)	Current Density at Peak EQE (mA cm ⁻²)	Year /ref
Nd-doped GaN	1082,1364	~50	~150	-	-	2004 ¹
Erbium tris(8-hydroxyquinoline)	1533	33	12	N/A	-	1999 ²
Rrbium organic compound	1540	~100	>10	N/A	<3	2000 ³
Conjugated polymer-Yb porphyrin blends	977	N/A	4	0.001	<200	2001 ⁴
Conjugation Platinum(II) Porphyrins	1005	>200	8.3	0.12	8-10	2011 ⁵
Selena-diazole containing polymers	990	>200	23.5	0.018	~20	2015 ⁶
D-p-A-p-D NIR chromophores	1050	220	4.8	0.33	0.1	2009 ⁷
D-A-D conjugated polymer	970	~150	1.1	0.05	<1	2004 ⁸
Single-walled carbon nanotubes	1177	50	2.1	0.01	<20	2018 ⁹
Y6	1040	~200	1.07	0.25	~10-100	2022 ¹⁰
Y11	1035		1.01	0.33		
IDSe-Th	1015		1.08	0.054		
Y-2W-F	1104		1.07	0.006		
SiOTIC-4F	1120		0.91	0.031		
COTIC-4F	1200		0.78	0.031		
TADF	1010	>250	3	0.003	20	2020 ¹¹
Yb(DBM) ₃ (H ₂ O) ₂ : DPE PO	1190	-	5-6	0.15	1	2017 ¹²
ErQ ₃ Singlet Fission OLED	1530	80	5	-	-	2018 ¹³
PbS in MEH-PPV	1160	190	5-6	0.27	-	2020 ¹⁴
InAs-ZnSe in MEH-PPV	1300	200	2.0	0.5	-	2002 ¹⁵
PbSe in MEH-PPV	1280	130	3.0	0.83	30	2003 ¹⁶

PbS QDs	1054	120	0.7-2.1	2.00	4	2012 ¹⁷
PbS-CdS QDs	1240	180	1.4	4.3	13.85	2015 ¹⁸
PbS in Perovskite	1391	150	<1	5.2	7.4	2016 ¹⁹
PbS QD-in-QD	1400	150	0.6	7.9	1-2	2019 ²⁰
PbS QD-in-QD	1400	150	0.7	8	200-400	2020 ²¹

References

- Kim, J. H. & Holloway, P. H. Near-infrared electroluminescence at room temperature from neodymium-doped gallium nitride thin films. *Applied Physics Letters* 85, 1689-1691, (2004).
- Curry, R. J. & Gillin, W. P. 1.54 μm electroluminescence from erbium (III) tris(8-hydroxyquinoline) (ErQ)-based organic light-emitting diodes. *Applied Physics Letters* 75, 1380-1382, (1999).
- Sun, R. G., Wang, Y. Z., Zheng, Q. B., Zhang, H. J. & Epstein, A. J. 1.54 μm infrared photoluminescence and electroluminescence from an erbium organic compound. *Journal of Applied Physics* 87, 7589-7591, (2000).
- Harrison, B. S. et al. Near-infrared electroluminescence from conjugated polymer/lanthanide porphyrin blends. *Applied Physics Letters* 79, 3770-3772, (2001).
- Graham, K. R. et al. Extended Conjugation Platinum(II) Porphyrins for use in Near-Infrared Emitting Organic Light Emitting Diodes. *Chemistry of Materials* 23, 5305-5312, (2011).
- Tregnago, G., Steckler, T. T., Fenwick, O., Andersson, M. R. & Cacialli, F. Thia- and seleno-diazole containing polymers for near-infrared light-emitting diodes. *Journal of Materials Chemistry C* 3, 2792-2797, (2015).
- Qian, G. et al. Simple and Efficient Near-Infrared Organic Chromophores for Light-Emitting Diodes with Single Electroluminescent Emission above 1000 nm. *Advanced Materials* 21, 111-116, (2009).
- Chen, M. et al. 1 micron wavelength photo- and electroluminescence from a conjugated polymer. *Applied Physics Letters* 84, 3570-3572, (2004).
- Graf, A., Murawski, C., Zakharko, Y., Zaumseil, J. & Gather, M. C. Infrared Organic Light-Emitting Diodes with Carbon Nanotube Emitters. *Advanced Materials* 30, 1706711, (2018).
- Xie, Y. et al. Bright short-wavelength infrared organic light-emitting devices. *Nature Photonics* 16, 752-761, (2022).
- Liang, Q., Xu, J., Xue, J. & Qiao, J. Near-infrared-II thermally activated delayed fluorescence organic light-emitting diodes. *Chemical Communications* 56, 8988-8991, (2020).
- Jinnai, K., Kabe, R. & Adachi, C. A near-infrared organic light-emitting diode based on an Yb(iii) complex synthesized by vacuum co-deposition. *Chemical Communications* 53, 5457-5460, (2017).
- Nagata, R., Nakanotani, H., Potscavage Jr, W. J. & Adachi, C. Exploiting Singlet Fission in Organic Light-Emitting Diodes. *Advanced Materials* 30, 1801484, (2018).
- Konstantatos, G., Huang, C., Levina, L., Lu, Z. & Sargent, E. H. Efficient Infrared Electroluminescent Devices Using Solution-Processed Colloidal Quantum Dots. *Advanced Functional Materials* 15, 1865-1869, (2005).
- Tessler, N., Medvedev, V., Kazes, M., Kan, S. & Banin, U. Efficient Near-Infrared Polymer Nanocrystal Light-Emitting Diodes. *Science* 295, 1506-1508, (2002).
- Steckel, J. S., Coe-Sullivan, S., Bulović, V. & Bawendi, M. G. 1.3 μm to 1.55 μm Tunable Electroluminescence from PbSe Quantum Dots Embedded within an Organic Device. *Advanced*

Materials 15, 1862-1866, (2003).

17 Sun, L. et al. Bright infrared quantum-dot light-emitting diodes through inter-dot spacing control. *Nature Nanotechnology* 7, 369-373, (2012).

18 Supran, G. J. et al. High-Performance Shortwave-Infrared Light-Emitting Devices Using Core-Shell (PbS-CdS) Colloidal Quantum Dots. *Advanced Materials* 27, 1437-1442, (2015).

19 Gong, X. et al. Highly efficient quantum dot near-infrared light-emitting diodes. *Nature Photonics* 10, 253-257, (2016).

20 Pradhan, S. et al. High-efficiency colloidal quantum dot infrared light-emitting diodes via engineering at the supra-nanocrystalline level. *Nature Nanotechnology* 14, 72-79, (2019).

21 Pradhan, S., Dalmases, M., Baspinar, A.-B. & Konstantatos, G. Highly Efficient, Bright, and Stable Colloidal Quantum Dot Short-Wave Infrared Light-Emitting Diodes. *Advanced Functional Materials* 30, 2004445, (2020).

Minor comments:

1. In Figure 1C and Figure 2e, it is imperative to accurately label all correlated transitions of the lanthanides to ensure a comprehensive and clear depiction of the data.

We have labelled the correlated transitions in Figure 1c and Figure 2e.

2. In Figure 2c and 2d, it is essential to specify which Ln nanoparticles correspond to the FTIR spectra. Furthermore, comprehensive FTIR curves for all other LnNPs, both before and after ligand exchange, should be included in the Supplementary Information

Thank you for the kind suggestion. The LnNPs we used in Figure 2c are YbNPs, and we have revised in Figure 2 accordingly. The comprehensive FTIR spectra of all other LnNPs before and after ligand exchange is added in Supplementary Fig. 3.

3. Figure S6 should feature the standard hexagonal phase XRD of nanoparticles.

We have added the standard data of β -NaGdF₄ (JCPDS card 27-0699) in Figure S6. All the diffraction peaks from different LnNPs are indexed and well matched with the standard hexagonal phase β -NaGdF₄ data.

Supplementary Fig. 6 | XRD patterns of different LnNPs along with standard XRD peaks of β -NaGdF₄ (JCPDS 27-0699).

Manuscript Title: Molecular triplets turn on insulating lanthanide-doped nanoparticles for NIR-II light-emitting diodes

Referees' comments:

Referee #1 (Remarks to the Author):

I have read through the response letter, the new manuscript, and the new supplementary information file in detail. The authors are to be commended for the impressive suite of new experiments conducted since the previous submission, resulting in mechanistic elucidation and, ultimately, to significant device improvements including orders-of-magnitude increase in EQE. The low efficiency was my primary concern in the previous round, and the fact that they now present numbers that are on par with/better than existing systems is impressive – and at the same time addresses my worry that there was no reasonable path towards improvement. While I am not sure that one should expect innovations over time comparable to those shown by the perovskite system the authors mention, they do make a compelling case that future improvements can also be possible.

I feel the added comparisons to other existing systems make their arguments stronger (Suppl. Tables 1-3). I also appreciate the initial core-shell mechanistic studies and their discussion of the tradeoffs offered by shells, and further agree that this is a topic for future study and improvement. I therefore find the updated manuscript novel, interesting, and potentially impactful, and thus recommend it for publication.

We thank Referee #1 for the high praise of our work and recommendation for publication in Nature.

Referee #2 (Remarks to the Author):

Re: Molecular triplets turn on insulating lanthanide-doped nanoparticles..., by Zhongzheng Yu, Zunzhou Deng, Junzhi Ye, et al.

This revised version has addressed the substantial weakness with the initial manuscript, notably the very low quantum efficiency. The core-shell Yb/Nd nanoparticles exhibit up to 0.6% (including an hemispherical outcoupling lens), which is now respectable if still far from the best QD devices.

In my view, the primary merit of the manuscript remains the interesting energy transfer scheme, where triplets are conveyed to the surface of the nanoparticles, and then some presumably find their way into the bulk. This scheme apparently solves the traditional problem with sensitizing rare earths - their inability to emit efficiently when directly coupled to an organic sensitizer. I suspect that many readers may find it frustrating that energy transfer within the nanoparticles largely remains obscure. Consequently, I'd encourage the authors to present what data they have concerning the difference between PL and EL (especially if the pump wavelength is such as to compare spectral differences when exciting the bulk directly versus via the surface). There are clearly important differences between PL and EL e.g. Fig. S25, but these pass unnoted by the

authors. Also, the efficiency improvement looks larger (100X?) than the PL difference (10X?). I think this merits a discussion of the potential importance of energy transfer away from the NP surface.

We thank Referee #2 for confirming that our revision has solved the main concern of the low external quantum efficiencies (EQEs) of these new type LnLEDs raised by both Referee #1 and Referee #2 in the previous round. We also appreciate that Referee #2 finds our energy transfer scheme interesting and effective.

To enhance the clarity, we have added the simplified energy transfer diagram within the Yb@Nd core-shell NPs in Supplementary Fig. 25 and shown below.

Extended Data Fig. 4c, Simplified energy transfer diagram from 9-ACA to Nd³⁺ to Yb³⁺ via surface excitation and from Nd³⁺ to Yb³⁺ via direct excitation of Nd³⁺.

We appreciate that Referee #2 point out the spectral differences between PL and EL. We find the disparity between PL and EL partially comes from batch-to-batch vibration. We have now re-measured, normalized and updated the PL spectra of Yb@Nd@9-ACA nanohybrids used to fabricate the best performance LnLEDs, at different excitation wavelengths, including 350 and 375 nm to excite the surface and 794 nm to directly excite the core-shell LnNPs, as shown in Supplementary Fig. 25d. Interestingly, the data showed spectral change when exciting the surface and the core-shell NPs. The Nd/Yb peak intensity ratio changed from 0.28 to 0.22, when switching from exciting the surface to the bulk of core-shell NPs in nanohybrids.

Extended Data Fig. 4d, Normalized PL spectra of Yb@Nd@9-ACA nanohybrids under

different excitation wavelengths at 350, 375 and 794 nm.

We have now added more discussions in the manuscript to highlight the spectral differences, in which EL spectra shows a higher Nd/Yb peak intensity ratio of 0.29 than the PL spectra of 0.22 by the direct excitation of Yb@Nd NPs. This indicates that electrical excitation prefers to active surface Ln³⁺ ions, and the energy transfer from Nd³⁺ to Yb³⁺ is less efficient than that under light excitation.

We would like to note that the EQE enhancement is a collaborative result of modifying the device structure and utilizing core-shell structure. Using the same device structure (TmPyPB and p-TPD), the Yb@Nd core-shell NPs has led to a 13.1-fold enhancement of EQE, compared to Nd-only LEDs.

A related concern is that the core-shell work is treated almost as an appendage even though its introduction has resulted in a ~100X improvement. I would consider updating the entire manuscript, starting with the device and energy transfer schematics in Fig. 1. The existing Fig. 4 is not especially interesting. 4c-f could probably sit comfortably in the supplementary. On the other hand, Fig. S25 should probably be in the main text.

We thank Referee #2 for this advice. As we stated above, the significant enhancement of the EQEs is the collaborative result of device structure optimization and core-shell structure design, especially for the low light-extraction efficiency in the NIR-II range. For the first report of these new type LnLEDs, we believe Fig. 1 represents the straightforward illustration of the design, efficacy, and mechanism of this complicated system even with the core-only LnNPs. We thus would like to assign our triplet energy concept to Fig. 1 and device optimization to Fig. 4, which we believe are both important for this manuscript.

Following the advice of Referee #2, we have modified Fig. 4 by removing Fig. 4d-f to Supplementary Information and adding parts of Supplementary Fig. 25 to Fig. 4.

The authors have included a table of comparable devices. While EQE is an important comparison point, the other crucial device characteristic that must be highlighted is the very low brightness of the devices. I'd expect that the QDs can get to much higher brightness if their emission is faster than the emissive states in the rare earths. Indeed, Figs. 4g-i show that the rare earth emission barely reaches 1mW/sr/m²), perhaps because the excited states are saturated? The rapid roll-off in the EQE shown in Fig. S25 f suggests that the core-shell nanoparticles are also hitting a low ceiling (although this is not directly quantified). Fortunately, the narrow emission of the rare earths is compatible with optical cavities that might speed up emission and at least partially address this issue. It still needs to be noted, however.

Thanks for the kind suggestion. We have highlighted in the manuscript that the brightness of the LnLEDs should be further studied and enhanced.

A few other minor comments:

- lines in the figures are frequently so thin as to be barely legible

We have increased the thickness of the lines in our Figures.

- Fig. 2e needs a legend

The legend of Fig. 2e now can be seen in the updated Fig. 2.

- line 268 of the supplementary should presumably refer to Fig. S23

Thanks for checking this and we have changed to Supplementary Fig. 19 after editorial formatting requirement.

Referee #3 (Remarks to the Author):

The authors have addressed some of my concerns; however, I still have concerns regarding the energy transfer study. A critical issue lies in the reported TET efficiencies, which are calculated to be 97.1%, 99.4%, and 98.5% for the Nd/Yb/ErNP@9-ACA nanohybrids. These values appear questionable given that the ligand exchange rate between 9-ACA and OA is reported to be only about 1%–6.8%, and that the emitters themselves are doped at only 20%. It seems unlikely that the small fraction of 9-ACA molecules bound to the surface could effectively interact with emitters and yield such high TET efficiencies. Upon closer examination, it appears that in the estimation of TET and throughout the spectroscopic studies, 9-ACA is consistently treated as the energy donor—perhaps contributing to the high TET efficiencies being reported. However, this assumption may not be entirely valid. Moreover, the core–shell design used for the LED device in the latter part of the work appears inconsistent with the main findings of the spectroscopic study. The doping strategy does not align with the conclusions drawn from the optical analysis, resulting in a disconnect between the fundamental photophysical insights and the final device architecture.

We thank Referee #3 for confirming that our revision has addressed the concerns raised in the previous round. We also appreciate the concerns raised by Referee #3 in this round to enhance the clarity of our manuscript.

We would like to stress that the TET efficiency is **not** related to the number of 9-ACA molecules bounded to LnNPs. The amount of bounded 9-ACA molecules would influence the number of absorbed photons, and consequently influence the overall emission intensity but not efficiency. The TET is a Dexter type energy transfer process, and the TET efficiency is sensitive to the donor-acceptor distance. From the DFT calculation, we can see that the 9-ACA molecules only bind to Ln³⁺ ions. The bounded 9-ACA molecules will undergo accelerated intersystem crossing to generated triplet excitons and then find the way to Ln³⁺ emitters even if they are bounded to inert Gd³⁺ ions.

To solve the concern that 9-ACA is treated as energy donor, we have added the data of GdNP@9-ACA nanohybrids as the reference, in which Gd³⁺ ions do not have energy levels available for energy transfer. A slightly accelerated decay of the triplet PIA of GdNP@9-ACA is observed compared to pure 9-ACA. Therefore, we have updated the **data and Methods** to avoid confusion. This is also consistent with our recent publication (van Turnhout, L. et al., Distance-Independent Efficiency of Triplet Energy Transfer from pi-Conjugated Organic Ligands to Lanthanide-Doped Nanoparticles. *J. Am. Chem. Soc.* **2024**, *146* (32), 22612-22621.).

The core-shell design did not show conflict with the spectroscopic study. In the core-shell structure, Nd³⁺ ions serve as the energy bridge between the triplet exciton of 9-ACA and Yb³⁺ ions to generate a high PLQE from Yb³⁺ for the core-shell thus require a high doping ratio.

We thank Referee #3 again for raising these concerns, but we hope that our reply could

clarify these misunderstandings and grant your final recommendation for publication in *Nature*.

My detailed comments are shown below:

1. In Fig. S5 and Fig. S6, why was the binding simulation conducted using Yb³⁺ or Y ions rather than Gd³⁺? Given that Gd³⁺ serves as the primary host ion in the nanoparticles, it would seem more appropriate to calculate the spectral shift based on Gd³⁺ coordination.

The simulation data of Gd³⁺ **has been shown** in the main Fig. 2c and d. We have included Yb³⁺ and Y³⁺ ions to show that this is a general method that works for different Ln³⁺ ions.

2. Why is the replacement ratio of Nd higher than that of Er, and significantly higher than that of Yb?

This is a good question. The sizes of Ln³⁺ ions contract from Nd³⁺ to Er³⁺ to Yb³⁺, and the Lewis acid harness increases. Oleic acid is a hard base, so binding strength is greater with Yb³⁺ than with Er³⁺ than with Nd³⁺. It becomes harder to replace with 9-ACA from Nd³⁺ to Er³⁺ to Yb³⁺ and ligand replacement ratio decreases.

Meanwhile, we agree that ligand exchange is difficult to monitor and study. We have stated in the manuscript that “*Ligand exchange is a dynamic process and can be influenced by numerous factors*” and this topic needs further study.

3. Why does the PL enhancement ratio in Figure 2 differ from the quantum efficiency enhancement ratio reported in Table S4? For instance, Nd-doped nanoparticles exhibit the highest quantum efficiency enhancement of 19.5-fold compared to Yb and Er. However, in the PL study, Nd shows the lowest enhancement, only 6.6-fold, which contradicts the trend observed in quantum efficiency.

The calculation of PLQE measurements involve the absorption, while the PL enhancement measurements require the same weight concentration. So, the comparison of enhancement should be only between the same type of LnNPs and their nanohybrids. In particular, the absorption of Nd³⁺ at 350 nm is stronger due to the energy levels of ⁴D_{1/2}, ⁴D_{5/2}, ⁴D_{3/2} to ⁴I_{9/2}, and the PL intensity of NdNPs is thus higher and the PL enhancement is the lowest with a 6.6-fold enhancement, but yields the highest PLQE enhancement when dividend by the absorbed photons.

4. The description of “PLQE measurements show that bound 9-ACA molecules on LnNPs have significantly decreased PLQE compared to pristine 9-ACA, consistent with the enhanced intersystem crossing (ISC) by coupling with LnNPs and efficient energy transfer to the Ln³⁺ ions (Supplementary Fig. 1)”, may be misleading. The reduced PLQE of bound 9-ACA is expected regardless of energy transfer, due to factors such as surface quenching or restricted molecular motion. Therefore, the decrease in PLQE cannot be unambiguously attributed to enhanced ISC or energy transfer to Ln³⁺ ions. In other words, the claim of ISC enhancement in the hybrid

nanoparticles should not be based on a comparison with pristine 9-ACA alone. Instead, it should be evaluated relative to 9-ACA bound to free Ln³⁺ ions or host nanoparticle NaGdF₄ without doping (energy transfer). A similar concern applies to the interpretation of the lifetime data in Fig. 3b, where the observed shortening is used to conclude “quenching of the excitations and pointing to ISC enhancement and energy transfer.” This issue also extends to the pump–probe spectroscopy in Fig. 3d and the calculation of TET efficiency.

Thanks for pointing this out. We have removed or revised our interpretation to avoid misleading. We would like to stress that the increased ISC rate of organic molecules by coupling LnNPs, including GdNPs, has been proven and reported in our previous publication (Han, S. et al., Lanthanide-doped inorganic nanoparticles turn molecular triplet excitons bright. *Nature* **2020**, 587 (7835), 594-599).

We do agree with the Refree#3 that the comparison should be with both pristine 9-ACA and GdNP@9-ACA nanohybrids. We thus have updated the reference using GdNP@9-ACA nanohybrids, in which Gd³⁺ ions do not have energy levels available for energy transfer. A slightly accelerated decay of the triplet PIA of GdNP@9-ACA is observed compared to pure 9-ACA. This is also consistent with our recent publication (van Turnhout, L. et al., Distance-Independent Efficiency of Triplet Energy Transfer from pi-Conjugated Organic Ligands to Lanthanide-Doped Nanoparticles. *J. Am. Chem. Soc.* **2024**, 146 (32), 22612-22621.). We have updated the Fig. 3d and corresponding calculation of TET efficiencies to avoid confusion.

Fig.3d, Kinetics of singlet decay, triplet growth and decay in pristine 9-ACA molecules and molecules attached to different types of LnNPs.

5. The core–shell nanoparticle design appears arbitrary and inconsistent with the conclusions drawn from the energy transfer efficiency analysis. In Supplementary Fig. S11, 50% Yb³⁺ doping

outperforms 20%, with the explanation that “increasing the doping ratio of Yb³⁺ significantly enhances the downconversion intensities for both YbNPs and YbNP@9-ACA nano hybrids, as cross-relaxation between Yb³⁺ ions is not a major loss channel, unlike in Er³⁺ and Nd³⁺.” This suggests that high Nd³⁺ doping is unfavorable due to cross-relaxation losses. However, the core-shell design adopts the opposite approach, using higher Nd³⁺ and low Yb³⁺ concentrations. This contradiction undermines the coherence between the spectroscopic findings and the structural design of the nano hybrids, making the optical study disconnected from the actual device architecture.

We would like to note that the design strategies for core and core-shell LnNPs are totally different. In the core-shell structure design, Nd³⁺ shell serves as an energy transfer layer, which should facilitate the bridging between triplet excitons and Yb³⁺ emitters to generate the high PLQE from Yb³⁺. Thus, it requires a high doping ratio of Nd³⁺ at 60% to accept more triplet excitons, instead of yielding bright emission itself. But for Nd-core only NPs, Nd³⁺ ions serve as both acceptor and emitter, where cross-relaxation is severe when energy cannot be rapidly transfer to Yb³⁺ ions. This strategy has been reported in previous publications (Zhang, Y. et al., Ultrasmall-Superbright Neodymium-Upconversion Nanoparticles via Energy Migration Manipulation and Lattice Modification: 808 nm-Activated Drug Release. *ACS Nano* **2017**, *11* (3), 2846-2857; Cao, C. et al., Energy Transfer Highway in Nd³⁺-Sensitized Nanoparticles for Efficient near-Infrared Bioimaging. *ACS Appl. Mater. Interfaces* **2017**, *9* (22), 18540-18548.)

Manuscript Title: Triplets electrically turn on insulating lanthanide-doped nanoparticles

Referees' comments:

Referee #1 (Remarks to the Author):

In the previous round of reviews, referees' raised valid points and questions regarding the mechanistic framework and related comparisons between different measurements, control studies, and improving clarity. In particular, I agree that an expanded comparison and presentation of PL and EL results, and the use of GdNP@9-ACA hybrids (and not just pristine 9-ACA) for quantitative comparison were both important suggestions. The comments also highlighted several descriptions in the manuscript that were susceptible to potential misunderstanding or ambiguity, and thus warranted further clarification. In my view, the responses and corresponding updates/changes in the manuscript and SI have nicely addressed these points (for example, see the updates to the new fig. 4 and fig. 3d), and have provided citations to relevant literature support where necessary. The work reveals several interesting directions that will benefit from further study by the larger community (and thus are outside the scope of the current report). I believe the exciting results (the (first-ever) demonstration of electrical pumping in these systems) and supporting evidence as presented here are suitable and worthy for publication.

We thank Referee #1 for the confirmation that our last revision has solved all the concerns from Referee #2 and #3, and recommendation for publication in *Nature*.

Referee #2 (Remarks to the Author):

I think the revision overall looks good, except that I am still concerned about the brightness. The authors now state 'Future experimental and theoretical work is also required to enhance the brightness and operational stability of LnLEDs...'

I am not sure this comment is really sufficient, and it also doesn't look like the brightness comparison was added to the table in the supplementary.

Also there is no analysis of the brightness limitations. For example, the current density that we might expect to saturate the quantum dots is:

$J_{\text{sat}} = q N d / t$, where q is the electron charge, N is the density of emissive sites, d is the thickness of the emissive layer, and t is the lifetime of the emitter.

In this work $d=15\text{nm}$. Thicker layers (as might be used in conventional applications of rare earth phosphor coatings) are challenging to implement because of the insulating character of the material.

The key question is what is N ? In particular, can individual quantum dots support multiple excitations?

Is there any experimental evidence that these dots can support multiple excitations at a time without loss? If not, at a dot volume of 1000 nm^3 , $N=1e18 \text{ cm}^{-3}$. Also assuming $t=1\text{ms}$ (I couldn't find transient data for Nd, Yb and Er dots in the manuscript).

These assumptions yield $J_{\text{sat}} = 0.2\text{mA/cm}^2$, which by eye looks *_very_* similar to the saturation that is experimentally observed in the devices Fig. 4i. These devices are behaving exactly as might be expected for materials that exhibit quenching for multiple excitations per quantum dot.

We thank Referee #2 for the advice to add more discussions about the brightness limitations of the LnLEDs, and we have added a general comparison with the brightness of QD LEDs in our manuscript.

For the brightness limitations and N (density of emissive sites), we believe that the Referee #2 made a mistake for the N calculation, as in our case the LnNP hosts multiple Ln^{3+} emitters per nanoparticle. We have now calculated and listed the correct number as shown below.

The volume of the spherical LnNPs with a diameter of 10 nm, in the form of $\text{NaGd}_{0.8}\text{F}_4:\text{Yb}_{0.2}$ is:

$$V = \frac{4}{3} \times \pi \times (5 \text{ nm})^3 = 523.6 \text{ nm}^3$$

The molar mass of NaGdF_4 is $258.25 \text{ g mol}^{-1}$, similar with NaNdF_4 of $243.22 \text{ g mol}^{-1}$, NaYbF_4 of $271.03 \text{ g mol}^{-1}$, and NaErF_4 of $266.24 \text{ g mol}^{-1}$. Therefore, we will use the molar mass of NaGdF_4 to estimate the N . The density ρ of NaGdF_4 is $\sim 5.65 \text{ g cm}^{-3}$, equal to $5.65 \times 10^{-21} \text{ g nm}^{-3}$.

The mass of a single NaGdF_4 NP is:

$$m = V \times \rho = 523.6 \times 5.65 \times 10^{-21} \approx 2.96 \times 10^{-18} \text{ g}$$

The moles of NaGdF_4 are:

$$n = \frac{m}{M} = \frac{2.96 \times 10^{-18}}{258.25} = 1.15 \times 10^{-20} \text{ mol}$$

Avogadro's number is 6.022×10^{23} . Number of Gd^{3+} ions per particle is as each formula unit contains 1 Gd^{3+} ion:

$$\text{Number of } \text{Gd}^{3+} \approx 1.15 \times 10^{-20} \times 6.022 \times 10^{23} \approx 6.93 \times 10^3$$

Gd³⁺ ion density in the nanoparticles (in cm⁻³) is calculated to be:

$$\frac{6.93 \times 10^3}{523.6 \times 10^{-21}} = 1.32 \times 10^{22} \text{ cm}^{-3}$$

Considering that the doping ratios of emitter ions Yb³⁺, Er³⁺ and Nd³⁺ ions are fixed at 20%, and the packed spheric LnNPs counts ~50% of the film volume. The final density of emissive sites will be:

$$N = 1.32 \times 10^{22} \times 20\% \times 50\% = 1.31 \times 10^{21} \text{ cm}^{-3}$$

This number is 3 orders of magnitude higher than the 1e18 cm⁻³ the reviewer mentioned. The emission of LnNPs is generated by localized ion excitation, thus individual LnNP can support multiple excitations.

The Ln³⁺ lifetimes of core LnNP@9-ACA nanohybrids are around 20 μs, the Yb³⁺ lifetime in Yb@Nd@9-ACA is 0.76 ms. Thus, the saturated radiative-recombination current density ($J_{r,sat}$) for core LnNPs is around:

$$J_{r,sat} = \frac{q N d}{t} = \frac{1.602 \times 10^{-19} \times 1.31 \times 10^{21} \times 10 \times 10^{-7}}{20 \times 10^{-6}} = 10.5 \text{ A cm}^{-2}$$

the $J_{r,sat}$ for core-shell LnNPs is around:

$$J_{r,sat} = \frac{q N d}{t} = \frac{1.602 \times 10^{-19} \times 1.31 \times 10^{21} \times 10 \times 10^{-7}}{0.76 \times 10^{-3}} = 276 \text{ mA cm}^{-2}$$

The calculated saturation $J_{r,sat}$ is 3-5 orders of magnitude higher than that of Referee #2 suggested, and higher than our measurement range. Besides, we also note that the injected charge carriers could be consumed by leakage due to a relatively low replacement ratio (<10%) of 9-ACA molecules on the surface of LnNP, and exciton quenching at higher voltages, as indicated by the efficiency droop (might to 0.001%). We believe the brightness of the current devices is not limited by the saturation of Ln³⁺ excited sates.

Referee #3 (Remarks to the Author):

The authors have addressed my question satisfactorily. I have no further concerns and therefore support the manuscript in its current form.

We thank Referee #3 for the recommendation for publication in *Nature*.